# Molecular architecture of the 90S small subunit pre-ribosome

**Qi Sun[1,2,3†], Xing Zhu[2†], Jia Qi[2,3,4†], Weidong An[2,3,5†], Pengfei Lan[3], Dan Tan[3], Rongchang Chen[2], Bing Wang[2,3], Sanduo Zheng[3], Cheng Zhang[3], Xining Chen[3], Wei Zhang[3], Jing Chen[1,2,3], Meng-Qiu Dong[3], Keqiong Ye[2,6*]**

[1]PTN Joint Graduate Program, School of Life Sciences, Tsinghua University, Beijing, China; [2]Key Laboratory of RNA Biology, Institute of Biophysics, CAS Center for Excellence in Biomacromolecules, Chinese Academy of Sciences, Beijing, China; [3]National Institute of Biological Sciences, Beijing, China; [4]Department of Biochemistry and Molecular Biology, College of Life Sciences, Beijing Normal University, Beijing, China; [5]College of Biological Sciences, China Agricultural University, Beijing, China; [6]University of Chinese Academy of Sciences, Beijing, China

**Abstract** Eukaryotic small ribosomal subunits are first assembled into 90S pre-ribosomes. The complete 90S is a gigantic complex with a molecular mass of approximately five megadaltons. Here, we report the nearly complete architecture of *Saccharomyces cerevisiae* 90S determined from three cryo-electron microscopy single particle reconstructions at 4.5 to 8.7 angstrom resolution. The majority of the density maps were modeled and assigned to specific RNA and protein components. The nascent ribosome is assembled into isolated native-like substructures that are stabilized by abundant assembly factors. The 5′ external transcribed spacer and U3 snoRNA nucleate a large subcomplex that scaffolds the nascent ribosome. U3 binds four sites of pre-rRNA, including a novel site on helix 27 but not the 3′ side of the central pseudoknot, and crucially organizes the 90S structure. The 90S model provides significant insight into the principle of small subunit assembly and the function of assembly factors.

*For correspondence:
yekeqiong@ibp.ac.cn

†These authors contributed equally to this work

**Competing interests:** The authors declare that no competing interests exist.

## Introduction

The small 40S and large 60S subunit (SSU and LSU) of eukaryotic ribosomes are assembled from four ribosomal RNAs (rRNAs) and 79 ribosomal proteins (r-proteins) in a highly complicated and dynamic process (*Henras et al., 2008*; *Kressler et al., 2010*; *Woolford and Baserga, 2013*). Numerous assembly factors (AFs) and small nucleolar RNAs (snoRNAs) associate with the two subunits to assist their maturation. Mutations in the genes of r-proteins and AFs have been found to cause various diseases (*Freed et al., 2010*).

The 18S rRNA in the SSU and the 5.8S and 25S rRNA in the LSU are co-transcribed into a long 35S precursor rRNA (pre-rRNA) in the nucleolus in the yeast *Saccharomyces cerevisiae* (Sc). The pre-rRNA also encodes four spacer sequences that need to be removed during processing. The 5′ region of pre-rRNA that consists of the 5′ external transcribed spacer (5′ ETS), 18S rRNA and the internal transcribed spacer 1 (ITS1) is co-transcriptionally packed into a terminal ball of ~40 nm in size (*Osheim et al., 2004*). These particles represent the earliest assembly intermediates of 40S and are known as the 90S pre-ribosome or the SSU processome (*Trapman et al., 1975*; *Dragon et al., 2002*; *Grandi et al., 2002*; *Phipps et al., 2011*). Within 90S, the pre-rRNA is cleaved at sites A0 and A1 in the 5′ ETS and site A2 in the ITS1, yielding a 20S pre-rRNA intermediate. The 20S pre-rRNA is

packed in the pre-40S particle and processed at site D in the cytoplasm to generate the mature 18S rRNA.

The 90S pre-ribosome is assembled progressively in a 5' to 3' order (*Chaker-Margot et al., 2015*; *Zhang et al., 2016b*). The 5' ETS associates with U3 snoRNA and 28 AFs into a 2.1 MDa particle. A subset of 5' ETS-associated factors exist as pre-assembled subcomplexes: UTPA, UTPB and U3 small nucleolar ribonucleoprotein (snoRNP) (*Watkins et al., 2000*; *Grandi et al., 2002*; *Krogan et al., 2004*). As the 18S rRNA region is transcribed, the U14 and snR30 snoRNAs and at least 37 AFs are recruited. The assembly of 90S is also highly dynamic. When the 18S rRNA is close to completion, U14 and snR30 snoRNAs and at least 14 protein factors assembled earlier to the 18S region begin to dissociate (*Zhang et al., 2016b*). The fully assembled 90S contains an unprocessed pre-18S rRNA, U3 snoRNA, approximately 51 assembly factors and 18 r-proteins with a molecular mass of 5.0 megadaltons (*Zhang et al., 2016b*).

Elucidating the molecular mechanism of ribosome assembly requires detailed structural information of pre-ribosomes at different assembly stages. Cryo-electron microscopy (cryo-EM) has been used to reveal structure for 60S pre-ribosomes (*Leidig et al., 2014*; *Greber et al., 2016*; *Barrio-Garcia et al., 2016*; *Wu et al., 2016*) and a late pre-40S particle (*Strunk et al., 2011*). These structures represent late assembly intermediates in which ribosomal subunits already take shape. Very recently, a cryo-EM structure of *Chaetomonum thermophilum* (Ct) 90S was determined at 7.3 Å (*Kornprobst et al., 2016*). However, the reported model was rather incomplete with many unmodeled and unassigned densities due to limited resolution. Here, we determined three cryo-EM maps of Sc 90S pre-ribosome with resolution from 4.5 to 8.7 Å. The majority of the maps have been modeled and assigned to specific RNA and protein component based on fitting of crystal structures, de novo model building, chemical crosslinking and mass spectrometry (CXMS) data. Our maps also contain many densities unseen in the Ct 90S map. The nearly complete 90S model provides major insights into the early assembly events of 40S subunit and the function of AFs.

## Results

### Structure determination

We affinity purified 90S from yeast using Noc4 fused to a C-terminal tandem affinity purification (TAP) tag as bait (*Grandi et al., 2002*; *Zhang et al., 2016b*). The purified Noc4-TAP sample contains a large number of proteins (*Figure 1—figure supplement 1A*). We first constructed a density map for the Noc4-TAP particle at approximately 25 Å by using negative stain EM and with the mature 40S structure as the initial model (*Figure 1—figure supplement 1B–E*). Then, we analyzed its structure with cryo-EM and obtained a density map with a resolution range of 6–20 Å and an overall resolution of 8.7 Å by the gold standard Fourier shell correlation (FSC) = 0.143 criterion (*Figure 1A and D*, *Figure 1—figure supplement 2*). RNA duplexes, protein α-helices and WD β-propeller structures were readily recognized in the density. However, the densities at the top and some peripheral regions are weak, probably due to the heterogeneity of Noc4-TAP particles that contain both unprocessed and processed pre-rRNAs (*Zhang et al., 2016b*). We next attempted to improve the sample homogeneity by genetic approach. The RNA helicases Dhr1 and Mtr4 function after the A0 cleavage of pre-rRNA (*de la Cruz et al., 1998*; *Colley et al., 2000*; *Sardana et al., 2015*). To stall ribosome maturation at certain stages, Dhr1 or Mtr4 was depleted and 90S was purified using Enp1-TAP as bait. Although Enp1 is present in both 90S and pre-40S particles (*Schafer et al., 2003*), the Enp1-TAP samples purified after Dhr1 or Mtr4 depletion contained mainly 90S particles, suggesting that 90S failed to develop into pre-40S. The two samples were subsequently analyzed by cryo-EM. Three density maps were constructed from the ΔDhr1 sample with resolution ranges of 6–20 Å and 7.8 to 9.5 Å overall resolution (*Figure 1B and D*, *Figure 1—figure supplement 3*). One map (state 1) is highly similar to but more complete than the Noc4-TAP map and the other two maps reveal different states of 90S and will be discussed elsewhere. The ΔMtr4 sample yielded a density map of a resolution range of 4–20 Å and an overall resolution of 4.5 Å that is also highly similar with the Noc4-TAP map (*Figure 1C and D*, *Figure 1—figure supplement 4*). Some bulky side chains of proteins are visible at the core region of the ΔMtr4 map (*Figure 1—figure supplement 5*). The three maps were essentially superimposable other than differences in resolution and local occupancy (*Figure 1—*

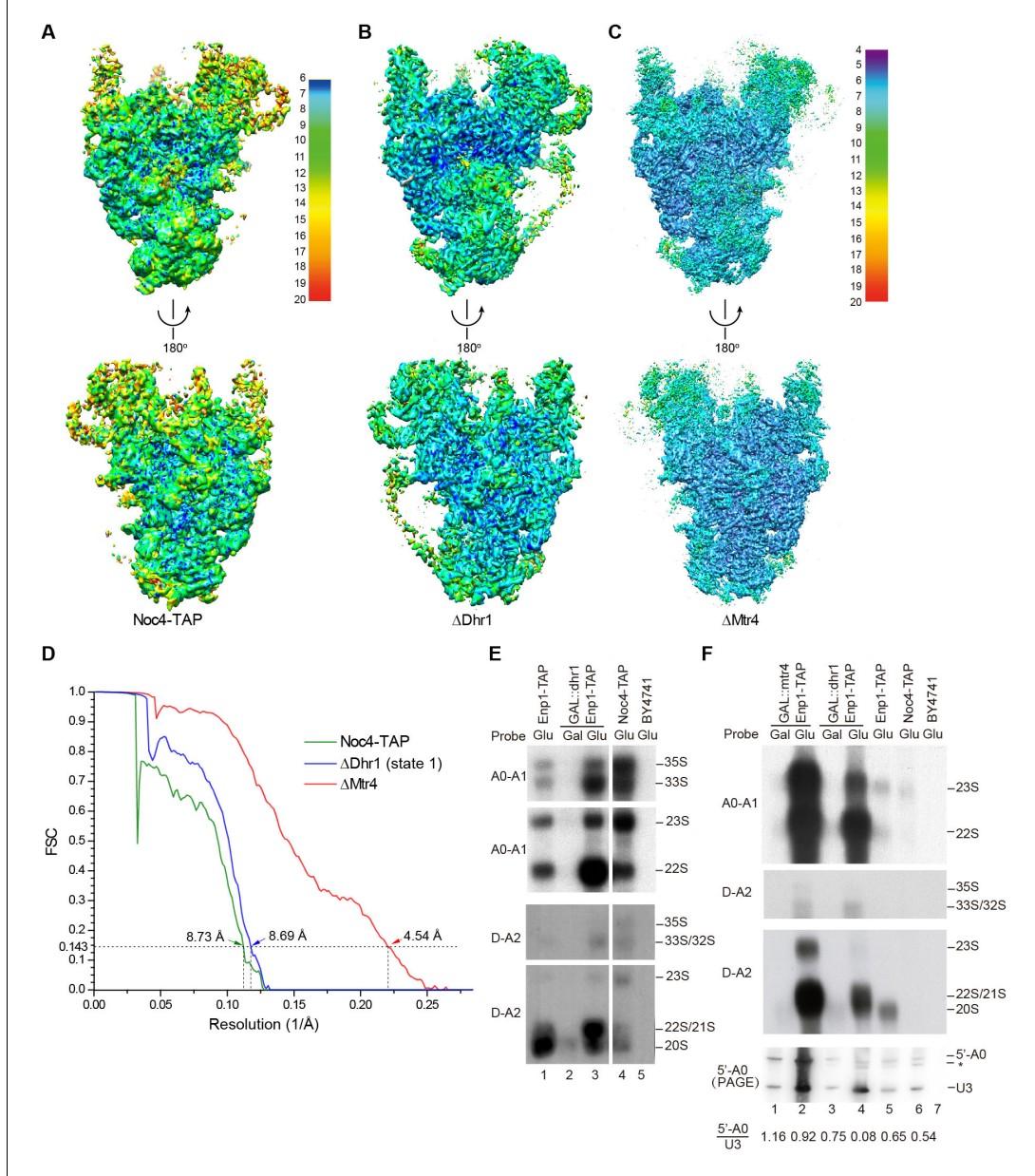

**Figure 1.** Cryo-EM analysis of 90S. (A–C) Local resolution cryo-EM maps of 90S derived from the Noc4-TAP (A), ΔDhr1 (state 1) (B) and ΔMtr4 (C) samples. The maps in A and B are colored with the same scale. Each map is shown in front and back views. (D) Gold-standard FSC curves for three constructions. (E–F) Northern blot analysis of RNA in purified pre-ribosomes. The GAL::dhr1/Enp1-TAP and GAL::mtr4/Enp1-TAP strains were grown in YPG (Gal) and shifted to YPD (Glu) for 14 hr to deplete the helicases. Pre-ribosomes were immunoprecipitated from equal $OD_{260}$ units of cell lysates and extracted for RNA. RNA was resolved in 1.2% agarose-formaldehyde gels or 8% polyacrylamide-8M urea gels (PAGE) and hybridized to $^{32}P$-labeled probes indicated. The wild-type cells have a much lower yield of pre-ribosomes as compared to the helicase-depleted cells. Two experiments are shown in E and F. The 35S and 33S RNAs were not efficiently transferred to membranes in F. Asterisk indicates a degradation product of the 5'-A0 fragment. The volume ratios of 5'-A0 fragment to U3 are calculated.

The following figure supplements are available for figure 1:

**Figure supplement 1.** Negative stain EM analysis of 90S samples.

**Figure supplement 2.** Cryo-EM analysis of Noc4-TAP 90S samples.

**Figure supplement 3.** Cryo-EM analysis of ΔDhr1 90S samples.

*Figure 1 continued on next page*

*Figure 1 continued*

**Figure supplement 4.** Cryo-EM analysis of ΔMtr4 90S samples.

**Figure supplement 5.** Representative densities fitted with structural models.

**Figure supplement 6.** Alignment of cryo-EM maps of 90S.

**Figure supplement 7.** Processing pathway of 18S rRNA.

*figure supplement 6A–B*). The initial Noc4-TAP map, the more complete ΔDhr1 map and the high-resolution ΔMtr4 map allowed us to build a more complete and accurate model.

The ΔMtr4 90S sample was also analyzed by chemical crosslinking and mass spectrometry (CXMS) (*Yang et al., 2012*). A total of 595 crosslinks, including 340 intermolecular and 255 intramolecular, provided rich structural information about 90S and greatly assisted the assignment of cryo-EM maps (*Supplementary file 2*).

The RNA species in purified pre-ribosomal particles were analyzed by northern blot (*Figure 1E–F*). The Noc4-TAP particle contains various processing intermediates: 35S, 33S, 23S, 22S and 20S pre-rRNAs (*Figure 1E*, lane 4; *Figure 1—figure supplement 7*). The Enp1-TAP sample purified from the wild-type yeast contains highly abundant 20S pre-rRNA and other intermediates present in 90S, as expected for the association of Enp1 with both 90S and pre-40S particles (*Figure 1E*, lane 1). On depletion of Dhr1 or Mtr4, the 22S pre-rRNA strongly accumulates in the Enp1-TAP sample (*Figure 1E*, lane 3; *Figure 1F*, lanes 2, 4), indicating that A0 cleavage is normal, but A1 cleavage and conversion of 90S to pre-40S are inhibited.

A0 cleavage produces a 5'-A0 fragment that was detected at comparable levels to U3 snoRNA in the wild-type and ΔMtr4 90S particles (*Figure 1F*, lanes 1–3, 5–6). This indicates that the 5'-A0 fragment still associates with 90S for a considerable period of time after A0 cleavage. Notably, the level of 5'-A0 fragment is significantly reduced (to ~10–20%) in the ΔDhr1 particle, apparently as a result of degradation or release. Consistent with the RNA analysis, the state 1 map of the ΔDhr1 sample has a weaker density for the 5' ETS RNA compared to the Noc4 and ΔMtr4 maps and the other two states are largely devoid of 5' ETS densities.

The EM densities were docked with the available crystal structures of yeast 40S ribosome (*Ben-Shem et al., 2011*) and AFs or homology models of AFs (*Figure 1—figure supplement 5*, *Supplementary file 1*). RNA duplexes were modeled for the 5' ETS and U3 RNA and poly-alanine chains (mainly α-helices) were de novo built for proteins without a structural model. The majority of the maps have been modeled and assigned (*Figure 2*, *Figure 3*, *Figure 3—figure supplement 1*). The model was refined against the density maps (*Supplementary file 3*) and includes 36 AFs, 19 r-proteins, 462 nucleotides (nt) of 5' ETS, 1103 nt of 18S, 12 nt of ITS1, 157 nt of U3 RNA and 1553 unassigned protein residues (*Supplementary file 1*, *Video 1*). The complete 90S particle contains approximately 51 stably associated AFs (*Zhang et al., 2016b*). The unidentified AFs and the missing region of assigned components could contribute to the unassigned densities or cannot be observed due to mobility or low occupancy.

## Overall structure

The 90S structure has a head shape with dimensions of approximately 300 Å x 310 Å x 360 Å (*Figure 2A*). Prominent structural features include two large standing ears, a small arch attached to the right ear, a large arch attached to the left ear, a nose in the front face, an open mouth, a jaw at the bottom and a small tail at the back (*Figure 2A*).

The bottom and back parts of 90S are made up by the 5' ETS subcomplex that accounts for about 40% of total mass (*Figure 2B–D*). UTPA, UTPB and U3 snoRNP are three large pre-assembled complexes associated with the 5' ETS. UTPA constitutes the jaw at the bottom. In addition, the UTPA protein Utp10 adopts an extended superhelical structure pointing upwards, forming the large arch. UTPB and U3 snoRNP locate at the right and left part of the back, respectively. The gap

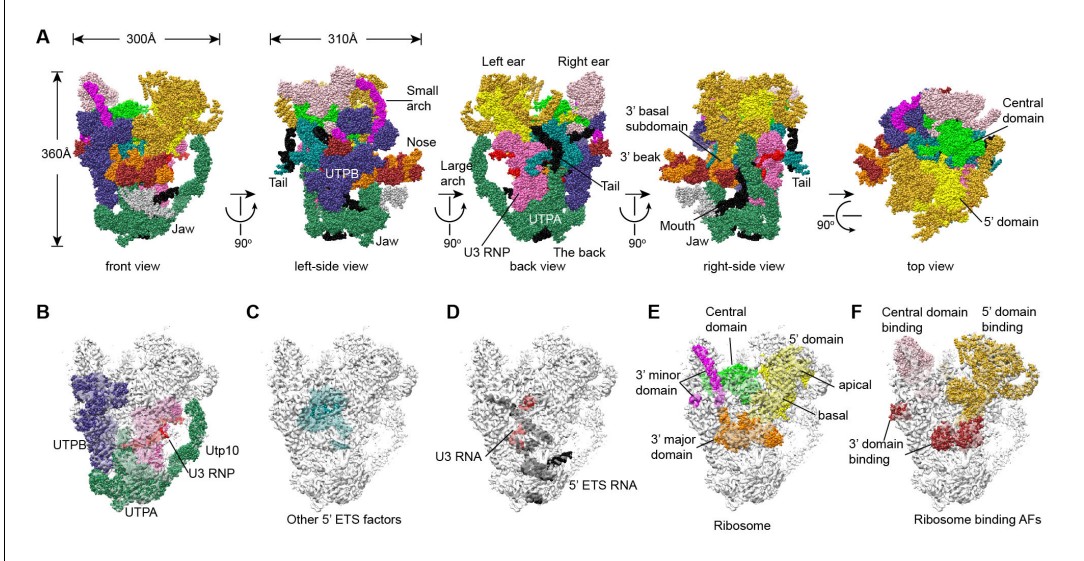

**Figure 2.** Structural organization of 90S. (**A**) Space-filling model of 90S in five perpendicular views. Substructures are color-coded as described below and other unassigned proteins are grey. Major structural features are indicated. (**B**) UTPA (sea green), UTPB (dark slate blue) and U3 snoRNP (hot pink). (**C**) Other 5′ ETS proteins including the unassigned proteins around 5′ ETS RNA (dark cyan). (**D**) 5′ ETS (black) and U3 (red) RNA. (**E**) The 5′ domain (yellow), central domain (green), 3′ major domain (orange) and 3′ minor domain (magenta) of small subunit ribosomes. (**F**) The AFs bound to the 5′ domain (goldenrod), central domain (pink), 3′ major domain (brown) and 3′ minor domain (brown). The substructures in B–F are shown in front view and with the ΔDhr1 density map.

between UTPB and U3 snoRNP is filled by UTPA proteins and other 5′ ETS factors, leading to a nearly seamless back. A small tail comprised of the 5′ ETS helices H8 and H9 and an unassigned protein (Unk2) projects out from the center of the back (*Figure 3B*).

The 18S rRNA is composed of the 5′ domain, the central domain, the 3′ major domain and the 3′ minor domain. These domains are partially assembled and placed at the top and front parts of the 90S structure (*Figure 2E*). The central domain and the apical part of 5′ domain, together with their associated AFs (*Figure 2F*), constitute the right and left ear, respectively. The long helix 44 in the 3′ minor domain forms a small arch attached to the right ear. The front face is made up by the basal part of 5′ domain, the basal part of 3′ major domain and the Bms1/Rcl1 complex. The beak subdomain of the 3′ major domain projects out from the front face, forming the nose. There is an open mouth beneath the nose, from which three long helices of 5′ ETS RNA stick out. The back and front faces enclose a large cavity that harbors the 5′ ETS and U3 RNA and their closely associated proteins.

## UTPB

UTPB is composed of Utp1/Pwp2, Utp21, Utp12/Dip2, Utp13, Utp8 and Utp6 (*Grandi et al., 2002*; *Krogan et al., 2004*). The first four proteins each contain tandem WD domains and form a tetramer by their homologous helical C-terminal domain (CTD) (*Zhang et al., 2016a*). UTPB adopts an elongated conformation in 90S, similar with the major conformation of the free complex (*Figure 4A*). The CTD tetramer is located at the center. The WD domains of Utp12 and Utp13, not interacting with each other, are situated at the up side of the CTDs. The WD domains of Utp1, Utp21 and Utp18 form a complex at the down side of the CTDs. The surface of the Utp21 and Utp1 WD domains are covered with some unmodeled densities that are likely from their binding partners (*Figure 1—figure supplement 5A*), such as the N-terminal region of Utp18 that binds at the Utp21 WD domain (*Zhang et al., 2014*). The HAT domain protein Utp6 that binds the N-terminal region of Utp18 was not located (*Zhang et al., 2016a*).

UTPB makes extensive interactions with UTPA, other 5′ ETS factors and the central, 3′ major and 3′ minor domain of the ribosome at the surface of 90S (*Figure 4E–F*, *Figure 4—figure supplement*

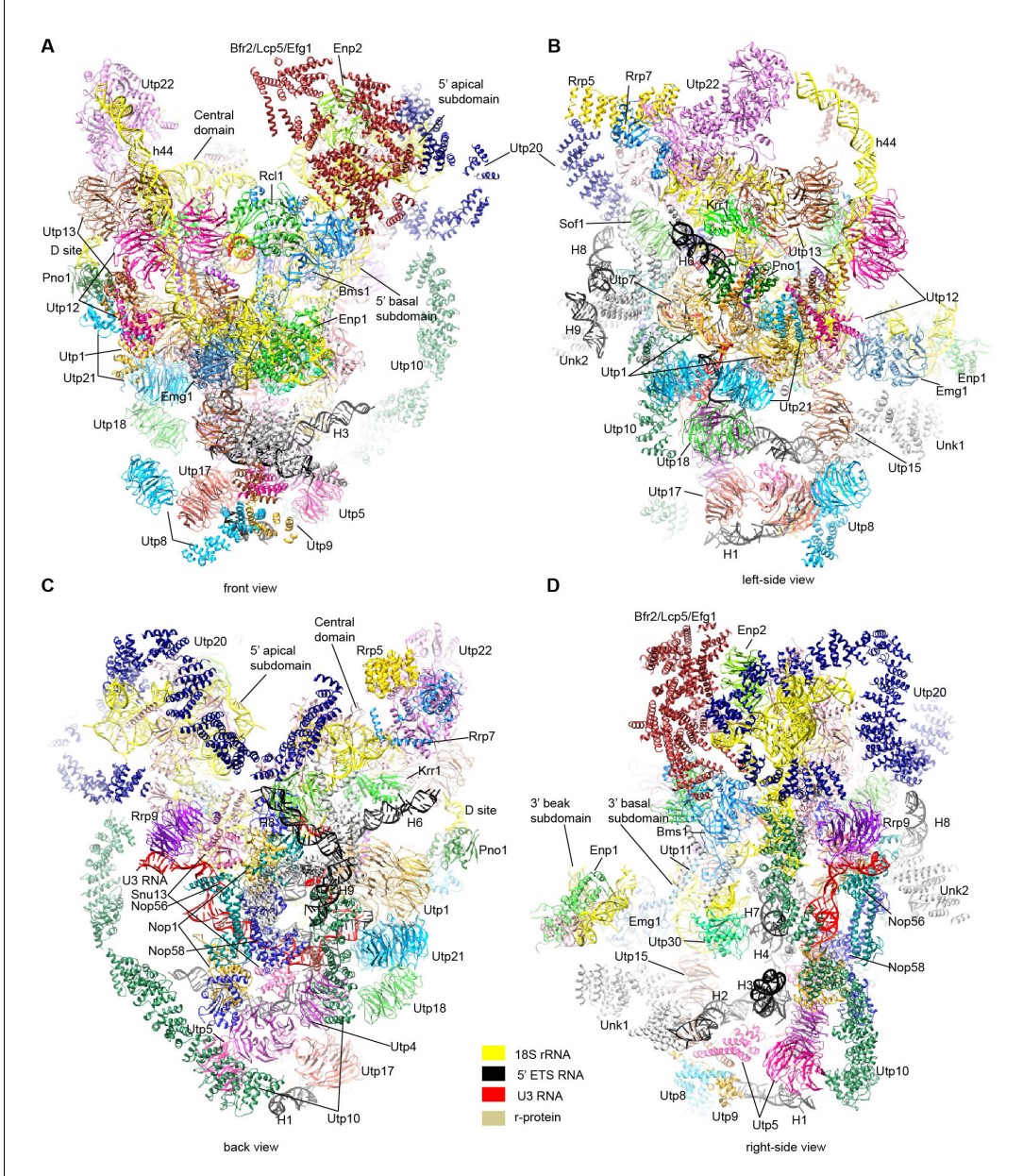

**Figure 3.** Structural model of 90S. (**A–D**) Structural model of 90S in the front (**A**), left-side (**B**), back (**C**) and right-side (**D**) view. The 18S, 5' ETS and U3 RNAs are colored yellow, black and red, respectively. The assigned AFs are color coded, all r-proteins are wheat and unassigned proteins are grey.

The following figure supplement is available for figure 3:

**Figure supplement 1.** Structural model of 90S.

*1*) and also binds the 5' ETS and U3 RNA located at the inside of 90S (Figure 7A). The WD domains of Utp18, Utp21 and Utp1 interact in order with the WD domain of Utp4, the N domain of Utp10 and the WD domain of Utp7, respectively (*Figure 4E*). The CTD tetramer and the WD domain of Utp1 contact the basal part of the 3' major domain (*Figure 4F*). At the top, the two projecting WD domains of Utp13 and Utp12 crucially organize the central domain, helix 44 and the D site RNA (*Figure 4F*). These interactions underscore the crucial role of UTPB components in 90S assembly (*Dosil and Bustelo, 2004*; *Pérez-Fernández et al., 2007*; *Pérez-Fernández et al., 2011*).

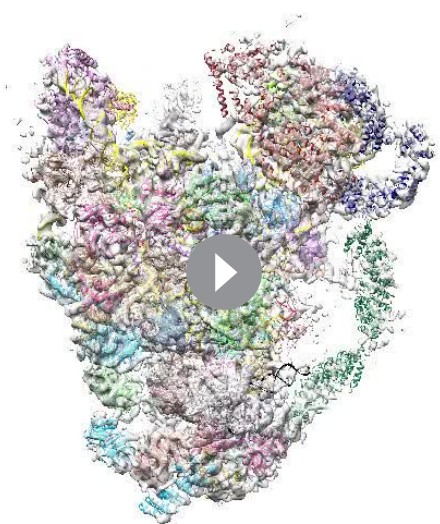

**Video 1.** Structure of 90S. The first two sections illustrate the ΔDhr1 (state 1) map and the structural model of 90S and the last section shows only the structure of nascent ribosome, 5' ETS and U3 RNA.

## UTPA

UTPA (aka t-Utp) is a large pre-assembled complex composed of Utp8, Utp9, Utp5, Utp15, Utp4, Utp17/Nan1 and Utp10 (*Gallagher et al., 2004*; *Krogan et al., 2004*). Domain and secondary structure prediction suggest that Utp8, Utp9, Utp5 and Utp15 each contain a WD domain and a helical C-terminal region and that Utp4 and Utp17 each contain tandem WD domains (*Figure 4B*). Two tandem WD domains and three WD domains identified at the jaw region were assigned to five UTPA proteins (*Figure 4D*). Utp9 appears to miss a WD domain.

We de novo built a helical structure at the jaw. Interestingly, the structure contains four subunits of similar fold and strikingly resembles with the CTD tetramer of UTPB (*Figure 4C*). The four subunits were assigned to the CTDs of Utp8, Utp9, Utp5 and Utp15 based on secondary structure analysis. Our structural data show that UTPA and UTPB are evolutionarily related and employ a homologous tetramerization domain for assembly. However, the four UTPA proteins contain a single WD domain, rather than tandem WD domains as in the UTPB proteins (*Figure 4D*).

Utp10 is a large protein (1769 residues) with mostly α-helices by prediction (*Dez et al., 2007*). The crystal structures of the N-terminal region (N domain) of Utp10 in complex with a C-terminal peptide of Utp17 and a middle region (M domain) of Utp10 were docked into two separated places in the density (PDB codes 5WYL, 5WY3), indicating that the N and M domains are linked by a disordered loop (*Figure 4D*). In the EM map, the M domain is continued by an elongated density that should correspond to the C-terminal region (C domain) of Utp10.

In addition to the tetrameric assembly of Utp5, Utp15, Utp8 and Utp9, the 90S structure also reveals a few interactions that probably mediate the assembly of UTPA proteins (*Figure 4D*). The WD domain of Utp5 contacts the WD domain of Utp4 and the M domain of Utp10. The WD domain of Utp17 bridges the CTD tetramer and the WD domain Utp4. The WD domain of Utp8 is attached on Utp17, forming a protrusion at the jaw. The N domain of Utp10 contacts with the WD domain of Utp4.

UTPA makes extensive interactions with the 5' ETS RNA (Figure 7) and supports the basal part of the ribosomal 3' major domain through Utp15 (*Figure 4F*). In addition, the WD domain of Utp4 and the N domain of Utp10, in an L-shape, wedge between the U3 snoRNP and UTPB complex and organize their positions in the back (*Figure 4E*).

## U3 snoRNP

U3 is a special box C/D snoRNA that does not function as a guide for 2'-O-methylation but plays an essential role in 90S assembly. U3 is divided into a 5' domain (nt 1–73) that base pairs with the pre-rRNA and a 3' domain (nt 74–333) that contains five helices P1 to P5 (*Figure 5A*) and associates with Nop56, Nop58, Snu13, Nop1 and Rrp9 (*Watkins et al., 2000*).

The U3 snoRNP structure shares a similar architecture with the monomeric assembly of an archaeal C/D RNP (*Lin et al., 2011*). Nop56 and Nop58, which are paralogues of archaeal Nop5, form a heterodimer through their coiled-coil domains (*Aittaleb et al., 2003*). The kink-turn structures formed by boxes C' and D and boxes B and C are sandwiched between Snu13 and the CTD of Nop58 and Nop56, respectively (*Cahill et al., 2002*). Two Nop1 molecules, in association with the N-terminal domain (NTD) of Nop56 and Nop58, are located at roughly symmetric places to the central coiled-coil domain.

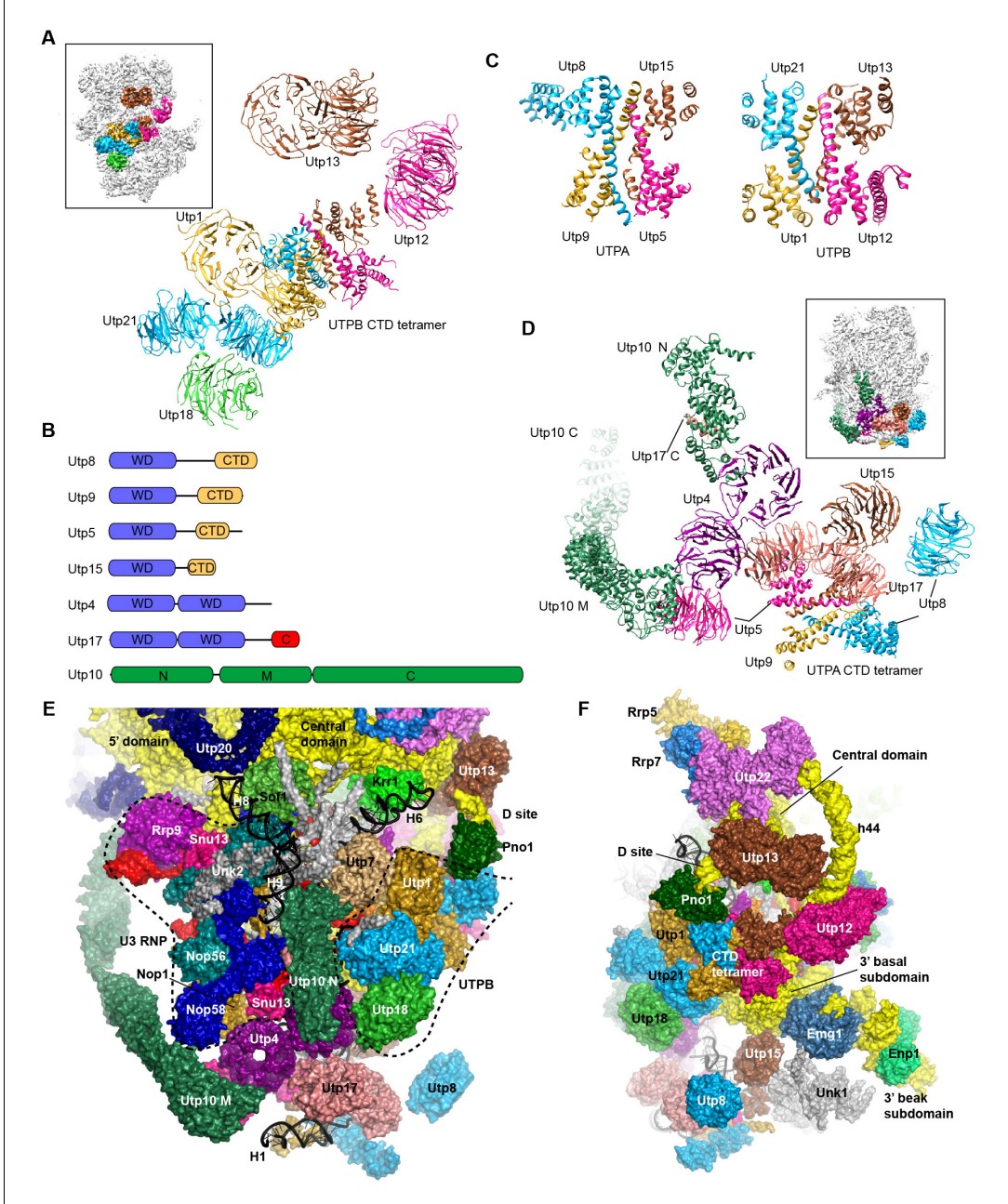

**Figure 4.** Structure of UTPB and UTPA. (**A**) Structure of UTPB in 90S. The insert shows the displayed region in the ΔDhr1 map. (**B**) Domain diagram of UTPA proteins. (**C**) UTPB and UTPA share a homologous CTD tetramer. (**D**) Structure of UTPA in 90S. The insert shows the displayed region in the ΔDhr1 map. (**E**) UTPA, UTPB, U3 snoRNP and additional 5' ETS factors form the back. UTPB and U3 snoRNP are marked with dotted lines. (**F**) Interface of UTPA and UTPB to the nascent ribosome.

The following figure supplement is available for figure 4:

**Figure supplement 1.** Structure of UTPB and UTPA.

Rrp9 is a U3-specific protein and does not associate with other C/D RNAs. The WD domain of Rrp9 contacts both Snu13 and the P4 helix of U3, which confirms the previous observations that Rrp9 binds near the box B/C region and Snu13 enhances the binding affinity and specificity of Rrp9 to U3 RNA (*Lukowiak et al., 2000*; *Granneman et al., 2002*; *Cléry et al., 2007*; *Zhang et al.,*

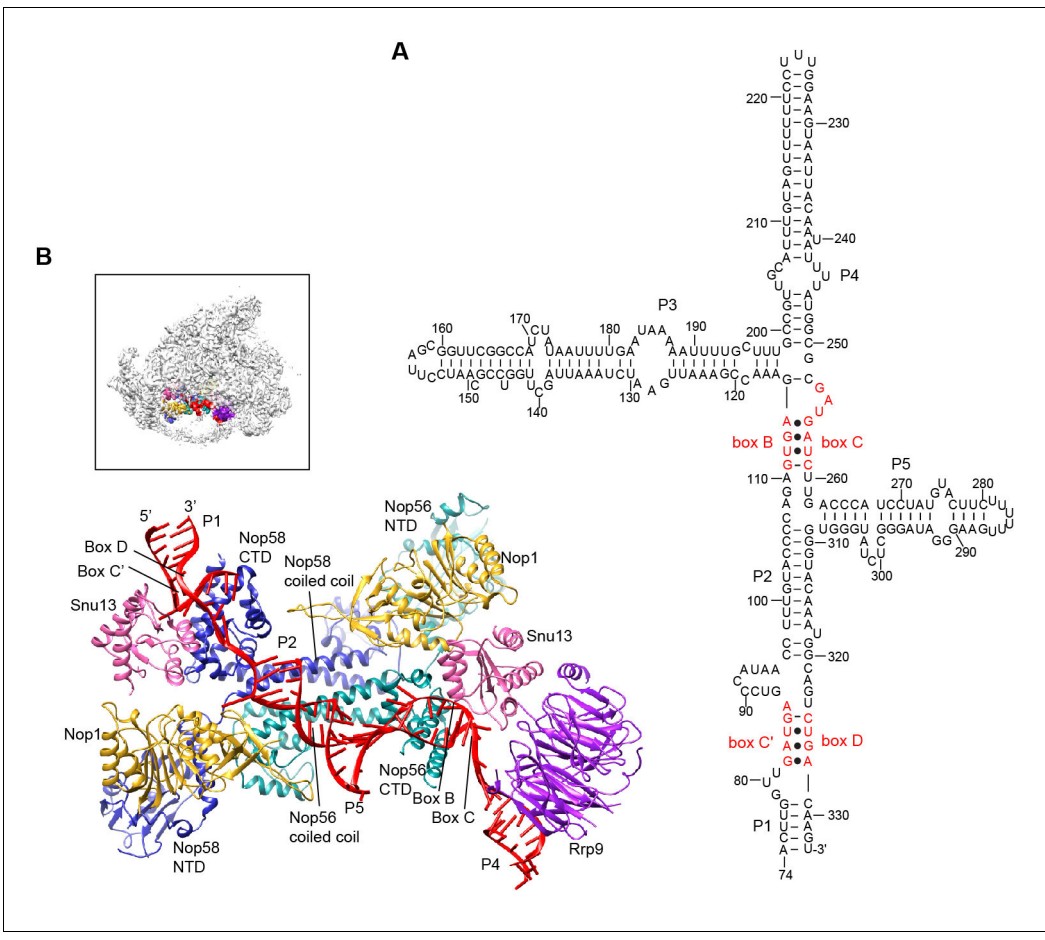

**Figure 5.** Structure of U3 snoRNP. (**A**) Secondary structure model of the 3' domain of U3 RNA. Paired regions are labeled as P1 to P5. The box C', B, C and D motifs are red. (**B**) Structure of U3 snoRNP in 90S. The insert shows the displayed region in the ΔDhr1 map.

The following figure supplement is available for figure 5:

**Figure supplement 1.** Structure of U3 snoRNP.

2013). The 7bc loop on the top face of the Rrp9 WD domain directly contacts Snu13, accounting for its important role in U3 binding (*Figure 5—figure supplement 1A*) (*Zhang et al., 2013*).

The two spacer sequences between two K-turns normally pair with substrates in box C/D guide RNAs, but they are predicted to pair with each other into helix P2 in the U3 RNA (*Figure 5A*). Interestingly, the P2 helix structurally resembles one guide-substrate duplex in the active archaeal C/D RNP structure (*Figure 5—figure supplement 1B–C*). The sequence between boxes D and C acts as a guide strand and the sequence between box C' and B as a substrate strand. The protruding α9' helix in the Nop5 CTD makes crucial interaction with the guide-substrate duplex (*Lin et al., 2011*). The corresponding α9' helix displays a well-ordered density in Nop58 as a result of interaction with P2, but is disordered in Nop56 in the absence of RNA interaction (*Figure 5—figure supplement 1D*). Despite the formation of a pseudo guide-substrate duplex, the methyltransferase Nop1 is not in the active conformation (*Figure 5—figure supplement 1B*). In addition, the CTDs of Nop56 and Nop58 are more separated compared to the active state of C/D RNP, representing a substrate-free relaxed state.

U3 snoRNP adopts a rectangular structure and assembles into 90S with U3 RNA facing inside (*Figure 3C*, *Figure 4E*, *Figure 4—figure supplement 1A*). This arrangement allows the 5' domain of U3 RNA penetrate into the 90S structure for binding the 5' ETS and 18S RNA. The box C'/D side of

U3 snoRNP (Nop1, Snu13 and the NTD of Nop58) fits into a U-shaped trench formed by Utp4 and Utp10. The box B/C side of U3 snoRNP (Rrp9, Snu13, Nop1 and the NTD of Nop56) contacts the basal part of the 5' domain of the ribosome, Sof1 and an unassigned protein Unk2.

## Other 5' ETS proteins

Besides UTPA, UTPB and U3 snoRNP, the 5' ETS RNA associates with Bud21, Utp7, Utp11, Mpp10, Imp3, Imp4, Sas10, Sof1, Fcf2 and Fcf1/Utp24 (*Chaker-Margot et al., 2015*; *Zhang et al., 2016b*). Imp3 and Imp4, bound to Mpp10 (*Lee and Baserga, 1999*), are buried inside the structure (Figure 7A–B). Utp24 is situated near the U3-18S duplex and the A1 site. The WD domains of Sof1 and Utp7 are located at the back. They and other unassigned densities between them seal the gap at the junction of U3 snoRNP, UTPA, UTPB and the central domain of the ribosome (*Figure 4E*). We de novo modeled two superhelical structures (Unk1, Unk2) (Figure 7F and I, *Figure 1—figure supplement 5G–H*) and many α-helices (Figure 7A–B). Many of the unassigned structures are close to the 5' ETS and U3 RNA and are likely from 5' ETS factors. A long α-helix that spans 104 Å was assigned to the N-terminal region of Utp11 by its crosslinking to Bms1 and S23 (Figures 7A–B and 9F).

## Structure of 5' ETS and U3 RNA

A secondary structure model has been previously proposed for the 5' ETS (*Yeh and Lee, 1992*). We modified the model to account for the number, length and connection of 5' ETS helices observed in the 90S structure (*Figure 6*). The new model includes 10 helices H1–H10. H3 and H7–H10 are largely preserved from the previous model, while other helices are considerably different. Eight helices (H1–H4, H6–H9) of 5' ETS were modeled into the EM map (*Figures 7C–D*, *Figure 1—figure supplement 5B*), while H5, H10 and its flanking sequences to the A0 and A1 sites were not located.

U3 snoRNA is predicted to bind two sites (nt 281–291, nt 470–479) of 5' ETS and the 5' end (nt 9–25) and central region (nt 1139–1143) of 18S rRNA (*Figure 6*, *Figure 6—figure supplement 1A*) (*Beltrame and Tollervey, 1992*; *Beltrame and Tollervey, 1995*; *Hughes, 1996*; *Sharma and*

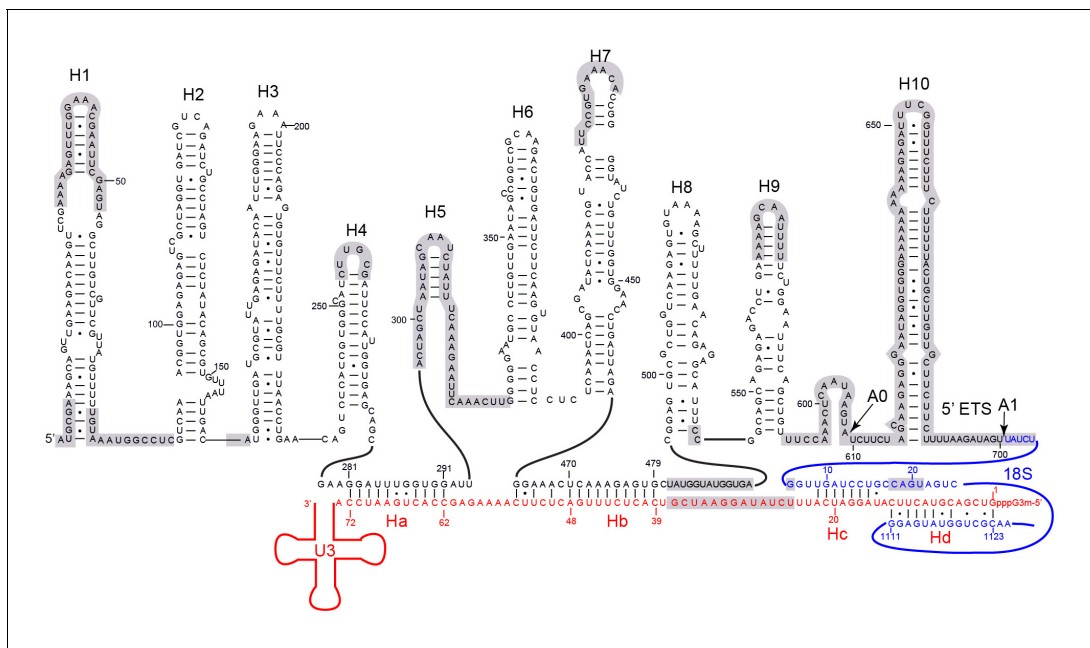

**Figure 6.** Secondary structure model of the 5' ETS and the 5' domain of U3 RNA. Paired regions of the 5' ETS RNA are labeled as H1 to H10. The 5' ETS and 18S RNA are numbered independently. Four hybrid duplexes of U3 observed in the 90S structure are named Ha, Hb, Hc and Hd. The grey-shaded sequences are not modeled.

The following figure supplement is available for figure 6:

**Figure supplement 1.** Interaction of U3 with 18S RNA.

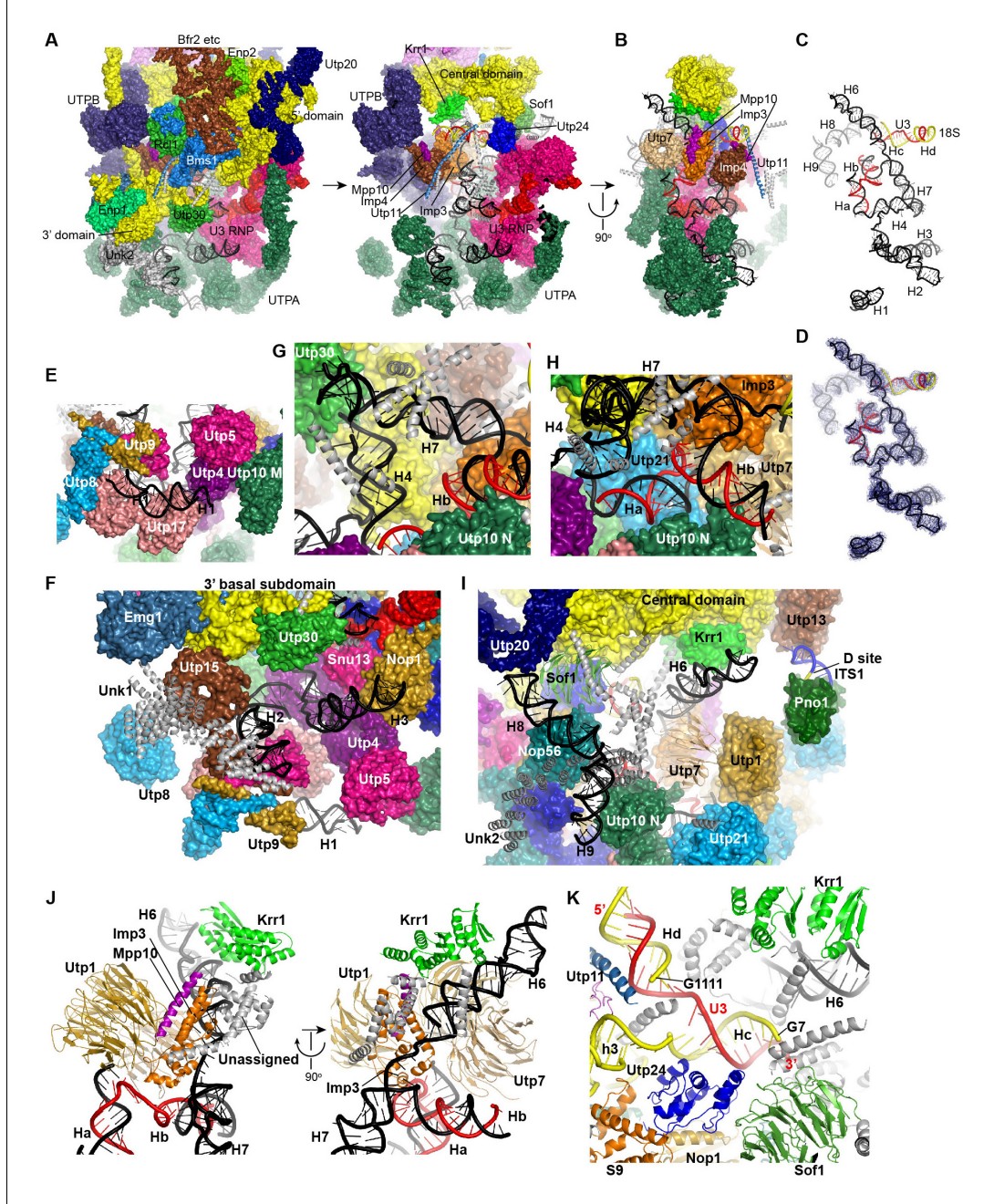

**Figure 7.** Structure and interaction of 5' ETS and the 5' domain of U3 RNA. (**A**) Open-box view of 5' ETS and U3 RNA. The left view shows an intact 90S structure. The right view is resulted from removal of the 5' and 3' domain of small subunit, Utp30, Enp1, Unk2, Utp20, Bms1, Rcl1, Enp2 and Bfr2. The unassigned structures are colored grey. (**B**) A 90° rotation of A. UTPB is additionally removed. (**C**) Structure of 5' ETS and the 5' domain of U3 RNA. Same orientation as in B. (**D**) The ΔMtr4 EM density for 5' ETS and the 5' domain of U3 RNA. (**E**) H1. (**F**) H2 and H3. A view of the mouth. (**G**) H4 and H7. (**H**) Ha and Hb. (**I**) H6, H8 and H9. (**J**) Imp3 binds at the junction of H6, H7, Ha and Hb. Two perpendicular views are shown. (**K**) Hc and Hd. The 5' nucleotides G7 and G1111 of the 18S strands in Hc and Hd are labeled.

*Tollervey, 1999*; *Dutca et al., 2011*; *Marmier-Gourrier et al., 2011*). Our structure reveals that U3 forms two helices Ha and Hb with 5' ETS and Hb is longer than predicted. U3 pairs with nt 9–16 of 18S into helix Hc, which is shorter than predicted (*Hughes, 1996*). Nucleotides 19–25 of 18S do not pair with U3 but form helix 3 in the 5' domain. The predicted interaction of U3 with nt 1139–1143 of 18S was not detected (*Hughes, 1996*).

Surprisingly, we found that the 5' end region (nt 1–13) of U3 pairs with an unknown sequence into a helix, termed Hd, adjacent to Hc (*Figure 6*, *Figure 7C*, *Figure 7K*, *Figure 1—figure supplement 5B*). Inspection of the unmodeled 18S RNA sequences suggest that the 5' arm of helix 27 (nt 1111–1123) could base pair with the U3 sequence into a 13-bp duplex that includes 2 mismatches and four wobble pairs. By forming the Hd helix with U3, the helix 27 sequence is positioned between the distant helices 26 and 28, consistent with their physical linkage (*Figure 8C*). In addition, this novel U3-18S interaction is conserved from yeast to humans (*Figure 6—figure supplement 1B*), which further supports the assignment. The U3-18S binding model is also consistent with the in vivo chemical probing result of U3 snoRNA (*Méreau et al., 1997*).

### Assembly of 5' ETS and U3 RNA

The 5' ETS RNA adopts a highly branched structure, in contrast to the compact and complicated fold of 18S RNA, and contacts mainly 5' ETS factors (*Figure 7A–D*). H1, H8 and H9 are located at the surface of 90S structure, H2, H3, H6 and H7 point their tips to the surface, while H4 and four U3 hybrid helices are deeply buried (*Figure 7A–D*).

The 5' region of 5' ETS (H1-H4) was shown to pull down and crosslink the UTPA complex in yeast (*Hunziker et al., 2016*; *Zhang et al., 2016b*). Consistently, this region primarily contacts the UTPA proteins. The exposed H1 lies over the tandem WD domains of Utp17 and contacts the CTD of Utp9 by its base and the M domain of Utp10 by its tip (*Figure 7E*). Along its outward path, H2 contacts the WD domain of Utp15, the CTDs of Utp15 and Utp5 and an unassigned superhelical structure

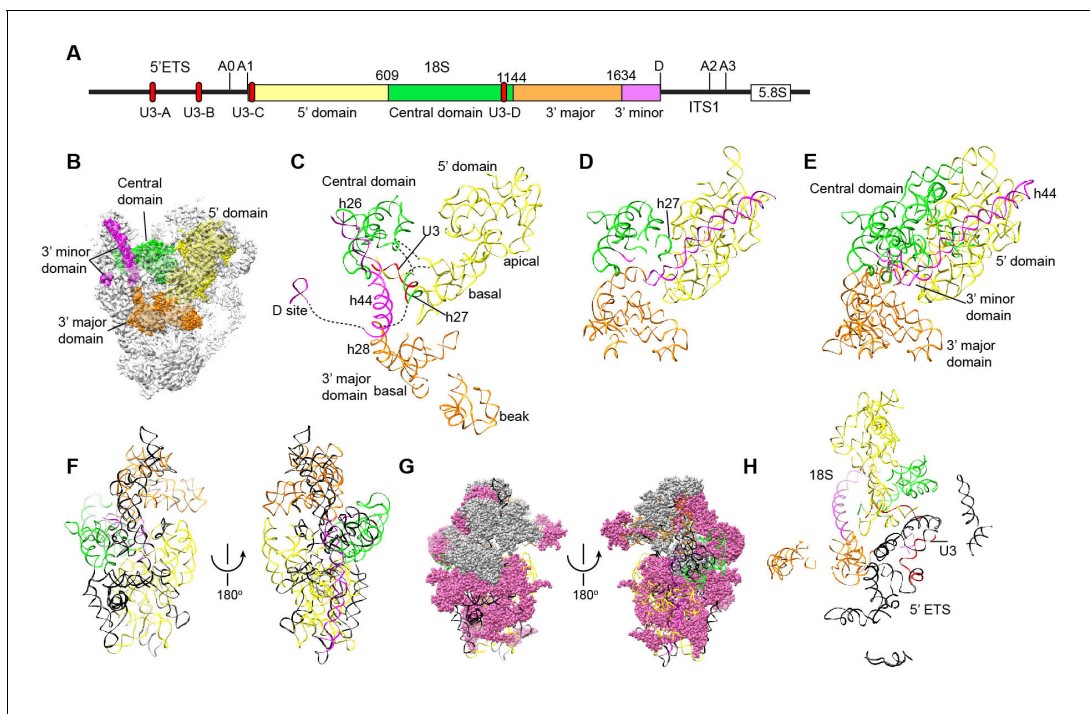

**Figure 8.** Nascent ribosome in 90S. (**A**) Diagram of pre-rRNA. 25S RNA is omitted. Processing sites, four 18S domains and four U3 binding sites are labeled. (**B**) Assembled ribosome in 90S. The 5' domain, central domain, 3' major domain and 3' minor domain of 18S and their associated r-proteins are colored in yellow, green, orange and magenta, respectively. The color theme is also used in other panels. (**C**) Structure of 18S RNA in 90S. Dotted lines show connections between major domains. (**D**) The 18S RNA detected in 90S is mapped to 40S structure. The central domain is aligned to the structure in both panels B and C. (**E**) 18S RNA in 40S structure. Same view as in **D**. (**F–G**) The 18S RNA (**F**) and r-proteins (**G**) observed in 90S are mapped to 40S structure. Undetected RNAs and proteins are colored black and grey. Detected r-proteins are colored deep pink. (**H**) Structural separation between the 18S and 5' ETS RNA.

The following figure supplement is available for figure 8:

**Figure supplement 1.** Assembly of 18S RNA in 90S.

Unk1 (*Figure 7F*, *Figure 1—figure supplement 5H*). Unk1 associates with Utp15 and Emg1 (*Figure 7F*). The base part of H3 is bent and contacts the tandem WD domain of Utp4 and Snu13. The top portion of H3 is attached to Nop1 through an unmodeled density (*Figure 1—figure supplement 5C*). H4 and H7, aligned in a right angle, are bound by a few unassigned α-helices (*Figure 7G*). The tip of H7 points to the mouth, whereas H6 sticks out from the back in the opposite direction of H7. The U3-5' ETS helix Ha interacts with the WD domain of Utp21 and the N domain of Utp10 (*Figure 7H*). Hb is perpendicular to Ha and binds the WD domain of Utp7.

Imp3, in association with a short peptide of Mpp10 (PDB code 5WXM) (*Lee and Baserga, 1999*; *Gérczei and Correll, 2004*; *Zheng and Ye, 2014*), binds at where H6, H7, Ha and Hb meet and critically organizes the junction structure (*Figure 7J*). Imp3 also contacts the WD domain of Utp1 and a few surrounding unassigned α-helices that extend to Krr1. The H6 helix is extensively bound by Imp3, unassigned α-helices, Krr1 and the WD domain of Utp7. The RNA sequence between Ha and Hb makes abundant interactions with 5' ETS factors, accounting for its essential role in assembling the 5' ETS subcomplex (*Zhang et al., 2016b*).

H8 and H9 are located at the back of the 90S structure and together with an unassigned superhelical structure Unk2 form the tail (*Figure 7I*, *Figure 1—figure supplement 5G*). Unk2 docks mainly on the NTD of Nop56 and forms a base for H8 and H9 attachment. H8 and H9 join their bases in about 120° and ride on Unk2. H8 extends to reach Utp20, whereas the upper stem of H9 has a weak density. The interaction of H8 and H9 with Unk2, which is probably a 5' ETS factor, would account for the observation that the assembly of 5' ETS subcomplex is stabilized by H8 and H9 (*Zhang et al., 2016b*).

The U3-18S helices Hc and Hd are located in a crowded region and surrounded by Sof1, Utp11, Utp24 and many unassigned α-helices (*Figure 7K*). The extensive protein interactions of the two U3-18S helices suggest their important role for 90S assembly. Consistent with their physical connection, the 3' end of the 18S strand in Hc is adjacent to helix 3 in the 5' domain. The 5' end of the 18S strand, which leads to the A1 site, is close to Utp24, supporting that Utp24 is the nuclease for site A1 (*Bleichert et al., 2006*; *Tomecki et al., 2015*; *Wells et al., 2016*).

## Nascent ribosome in 90S

The four domains of 18S RNA are assembled into several subdomains that adopt native-like structures, but are not yet packed (*Figure 8A–F*, *Figure 8—figure supplement 1*). The RNA elements at the domain/subdomain interface or the AF binding sites are frequently shifted from their native positions or disordered (*Figure 8F*). Despite lack of inter-domain packing, the partially assembled 5' and central domains already take a native orientation as in the mature 40S structure (*Figure 8C–D*). Such structural organization should facilitate their joining into the body at the next step of assembly. By contrast, the 3' major domain and the 3' minor domain are placed far away from their native positions (*Figure 8C–D*).

The 5' domain is better assembled compared with the central and 3' major domain. The latter two domains still contain a large fraction of disordered RNAs (*Figure 8C–F*, *Figure 8—figure supplement 1*). 19 out of 33 r-proteins in 40S ribosome were identified in the EM density (*Figure 8G*, *Supplementary file 1*). The missing r-proteins are located mainly at the head (S0, S3, S10, S15, S17, S18, S19, S20, S25, S29, Asc1) and the head-body junction (S2, S21). One protein S26 bound to the central domain is missing. The 18S RNA is well separated from the 5' ETS RNA, reflecting their functional and structural distinction (*Figure 8H*).

## The 5' domain

The 5' domain is assembled into a basal subdomain (helices 3–4, 15–18 and r-proteins S9, S23, S24 and S30) and an apical subdomain (helices 5–14 and r-proteins S4, S6, S8 and S11) and heavily covered by AFs (*Figure 9A–D*, *Figure 8—figure supplement 1*). The assembled 5' domain already takes a native-like overall shape, but the two subdomains are slightly twisted and loosely packed.

The 5' apical subdomain and its associated AFs stand out from the main body of 90S structure and form the left ear. A few RNA elements are not assembled into their final positions in the apical subdomain. Helix 6, the binding site for U14 snoRNA (*Liang and Fournier, 1995*), is disordered (*Figure 9C*). U14 transiently associates with the early assembly intermediates of 90S and is released

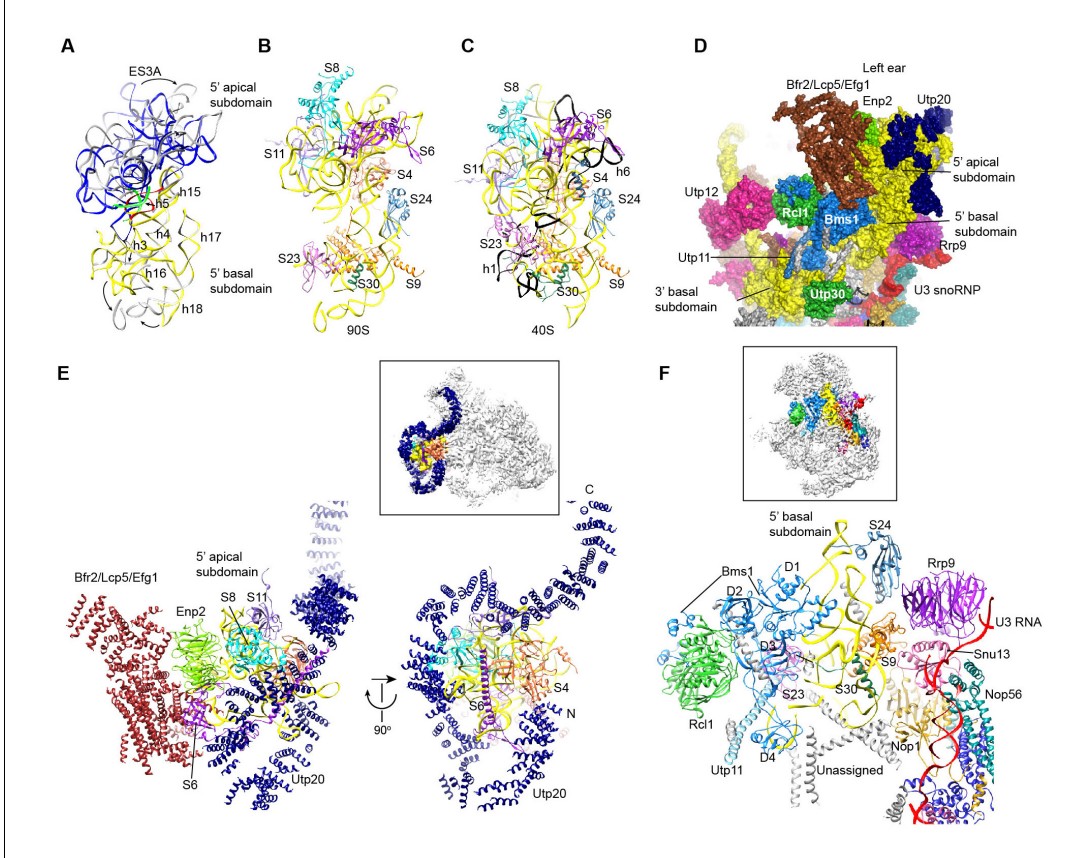

**Figure 9.** Structure of the 5' domain in 90S. (A) The 5' domain RNA in 90S (grey and green) and 40S (yellow, red and green) are aligned by the basal subdomain. Shifted RNA elements are marked with arrows. (B–C) The assembled 5' domain structure in the 90S (B) and 40S (C) ribosome. RNAs disordered in 90S are colored black in C. Same orientation as in A. (D) Structure of the 5' domain and associated AFs shown in surface view. The beak region is removed for clarity. (E) Interactions of the 5' apical subdomain with AFs are shown in two perpendicular views. The insert shows the displayed region in the ΔDhr1 map. (F) Interactions of the 5' basal subdomain with AFs. The insert shows the displayed region in the ΔDhr1 map.

in our samples that contain mainly complete 90S particles (*Zhang et al., 2016b*). The peripheral RNA helices ES3A and ES3B are shifted or disordered as a result of Utp20 binding.

A pre-rRNA fragment that terminates after the 5' apical subdomain (position 435) was shown to recruit Enp2, Bfr2, Lcp5, Efg1 and other unstable factors (*Zhang et al., 2016b*). Enp2 is assigned to the only WD domain that is located around the 5' apical subdomain and contacts S8 and helix 14 (*Figure 9E*). A large block of density is associated with the WD domain of Enp2 and the N-terminal domain of S6 (*Figure 1—figure supplement 5J*). The density shows a complex fold and probably comes from the rest part of Enp2, Bfr2, Lcp5 and Efg1.

Opposite to the Enp2-binding site, a very long density wraps around the 5' apical subdomain in about 3/4 circles and then extends to bind the central domain, forming a large S on the top of 90S structure (*Figure 3C–D*, *Figure 9E*, *Figure 1—figure supplement 5I*). A superhelix structure was de novo modeled into the density and assigned to Utp20. Utp20 is the longest protein (2493 residues) in 90S and contains helical HEAT repeats (*Dez et al., 2007*). Utp20 contacts both the 5' domain and the central domain and appears to stabilize their conformation in the partially assembled ribosome and facilitate their joining into the body as the ribosome assembly proceeds.

The 5' basal subdomain is buried in the main body of the 90S structure and contacted by Bms1, U3 snoRNP and a few unassigned α-helices from the structure interior (*Figure 9D and F*). Due to AF contacts, helix 3, its associated r-protein S23 and the tip of helices 16 and 18 are shifted (*Figure 9A–C*).

Bms1 is a GTPase and associates with Rcl1 (*Gelperin et al., 2001*; *Wegierski et al., 2001*; *Karbstein et al., 2005*). A structural model of Bms1 was built on the crystal structure of Tsr1 (PDB code 5WWN and [*McCaughan et al., 2016*]), which is a pre-40S assembly factor and homologous to Bms1(*Gelperin et al., 2001*). Bms1 consists of four domains (D1-D4) and interacts extensively with the 18S RNA. The D1 GTPase domain binds helices 3 and 4 and the D3 and D4 domains bind helix 18. The binding site of Bms1 overlaps significantly with that of Tsr1 (*Strunk et al., 2011*), suggesting that Bms1 is replaced by Tsr1 during the transition of 90S to pre-40S.

Rcl1, which binds a middle sequence (residues 547–636) of Bms1 (*Delprato et al., 2014*), is located next to Bms1 (*Figure 9D and F*). In addition, Rcl1 also contacts the D2 and D3 domain of Bms1. The Bms1 and Rcl1 complex is positioned at the front face of 90S structure and makes multiple interactions with other structural modules. Rcl1 docks on the WD domain of Utp12 (*Figure 9D*), whereas the D4 domain of Bms1 contacts the basal subdomain of the 3' major domain (*Figures 9D and 11I*). The long N-terminal α-helix of Utp11 bridges the D4 domain of Bms1 and S23 (*Figure 9F*).

## The central domain

The central domain, helix 45 and the 3' end of 18S RNA assemble with seven r-proteins (S1, S7, S13, S14, S22, S26 and S27) into the platform in the mature 40S structure. In 90S, the outskirt of the platform is assembled from helices 20, 22, 23 and 26 and six r-proteins except S26 (*Figure 10A–B*, *Figure 8—figure supplement 1*). A few RNA elements placed at the interface to the 5' domain and the 3' minor domain in the mature 40S structure are not ordered. In particular, the large eukaryote-specific expansion segment 6 (ES6) that makes long-range interactions with the 5' domain was not located (*Figures 8F* and *10B*).

Krr1, Utp22, Rrp7 and Rrp5, whose association depends on the central domain (*Chaker-Margot et al., 2015*; *Zhang et al., 2016b*), are indeed associated with the central domain. Krr1 binds at the inner side of the L-shaped helix 23 and also contacts r-proteins S1 and S14. The binding

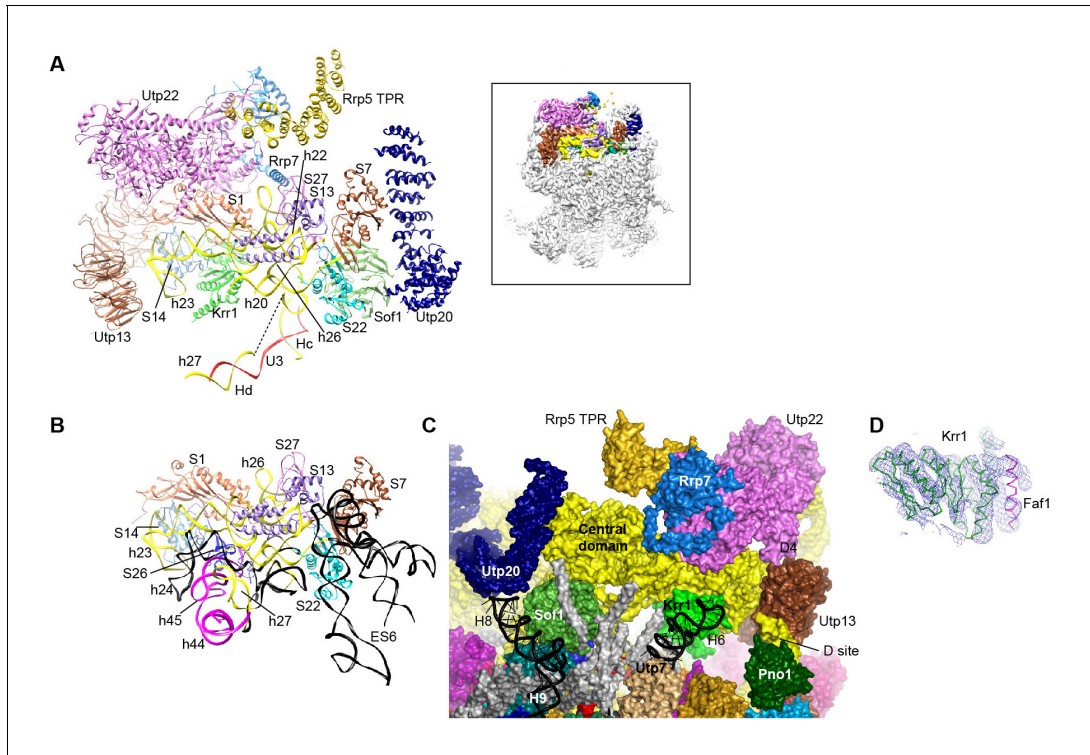

**Figure 10.** Structure of the central domain in 90S. (**A**) Structure of the central domain with bound AFs in 90S. The insert shows the displayed region in the ΔDhr1 map. (**B**) The central domain structure in the mature 40S is shown in the same view as **A**. The 18S RNA regions not detected in 90S are colored black. (**C**) Interface of the central domain with AFs in surface view. The orientation is nearly opposite to **A**. (**D**) Structure of the Krr1-Faf1 complex is fitted into the ΔDhr1 density map.

site of Krr1 significantly overlaps with that of helix 45 and S26. Consequently, helix 45 and S26 are occluded from the platform and the nearby helix 24 is also disordered (*Figure 10B*). Helix 27 is packed with helices 44 and 45 in the 40S structure. However, the 5' arm sequence of h27 is bound by U3 and placed midway between the central domain and the 3' major domain (*Figures 7K*, *8C* and *10A*). Krr1 is known to bind a short α-helix of Faf1 (*Karkusiewicz et al., 2004*; *Zheng et al., 2014*). However, no corresponding density was found for the Faf1 helix (*Figure 10D*), suggesting that the Faf1-Krr1 interaction is not populated in our samples.

The Utp22-Rrp7 complex is placed at the right ear of 90S structure (*Figure 10A*) (*Krogan et al., 2004*; *Lin et al., 2013*). Utp22 primarily associates with r-protein S1. The C-terminal tail of Rrp7 (residues 190–297) is required for 90S association, but is disordered in the crystal structure of the Utp22-Rrp7 complex. The EM map shows that part of the tail folds into α-helices and directly contacts the tip of helix 26 and S27. These interactions explain the essentiality of the tail and the previously observed crosslinking between Rrp7 and helix 26 (*Lin et al., 2013*).

An elongated density projects from Utp22 (*Figure 1—figure supplement 5L*). The density, although weak, was well fitted by the crystal structure of the C-terminal tetratricopeptide repeat (TPR) domain of Rrp5 (PDB code 5WWM and [*Khoshnevis et al., 2016*]). The interaction between Rrp5 and Utp22 is consistent with their similar assembly time on pre-rRNA (*Chaker-Margot et al., 2015*; *Zhang et al., 2016b*) and also provides a structural basis for the essential function of the C-terminal half of Rrp5 in 18S processing (*Eppens et al., 1999*). The 12 S1 RNA-binding domains of Rrp5 and their primary RNA binding target ITS1 were not located (*Lebaron et al., 2013*).

The central domain sits on the 5' ETS subcomplex with contacts to Sof1 and Utp13 (*Figure 10A and C*, *Figure 3C*). In addition, the C-terminal portion of Utp20 contacts Sof1 and S7, linking the central domain to the 5' domain.

## The 3' major domain

The 3' major domain associates with 18 r-proteins into the head of 40S ribosome. Only five of these r-proteins were located on the rather broken head in 90S. The 3' major domain is assembled into a basal subdomain, which is composed of helices 28–30 and 41–43 and r-proteins S5, S16 and S28, and a beak subdomain composed of helices 32–34 and r-proteins S12 and S31 (*Figure 11A–B*, *Figure 8—figure supplement 1*). The 3' basal subdomain makes up part of the front face, whereas the 3' beak subdomain sticks out to form the nose. The two subdomains are separated by Emg1 that binds to its modification target in the intervening helix 31 (*Figure 11C*, *Figure 1—figure supplement 5M*). Helix 34 points to a cluster of very weak densities that should correspond to the missing helices 35–40 (*Figure 1—figure supplement 5M*). Helix 42, the binding site for S18, is disordered (*Figure 11B*).

The 3' major domain RNA is directly bound by Enp1, Emg1, Utp30 and Imp4. The crystal structure of Enp1 was fitted to a density at the beak (PDB code 5WWO) (*Figure 1—figure supplement 5M*), consistent with its binding site at the pre-40S particle (*Granneman et al., 2010*; *Strunk et al., 2011*). Enp1 adopts a superhelix structure and contacts helices 32 and 33 (*Figure 11D*).

Emg1 is the methyltransferase for pseudouridine 1191 on the tip of helix 31(*Wurm et al., 2010*). The crystal structure of Emg1 dimer in complex with substrate RNA was docked into a density between the basal and beak subdomains (*Thomas et al., 2011*) (*Figure 11E*, *Figure 1—figure supplement 5M*). The Emg1 subunit loaded with the modification target further docks on S5 and the linker RNA between helices 29 and 42, stabilizing the association of Emg1.

Imp4 is a BRIX domain-containing protein and associates with Mpp10 (*Lee and Baserga, 1999*; *Wehner and Baserga, 2002*). The crystal structure of another BRIX domain protein Rpf2 in complex with Rrs1 (PDB code 5WXL and [*Asano et al., 2015*; *Kharde et al., 2015*; *Madru et al., 2015*]) was used as the structural model for Imp4. Imp4 is located at the interior of the 90S structure and extensively contacts helices 43 and 28 (*Figure 11F*). A short α-helix bound at the protein-binding pocket of Imp4 was tentatively assigned to Mpp10 based on the known interaction between Imp4 and Mpp10 (*Lee and Baserga, 1999*; *Wehner and Baserga, 2002*).

The top region of helix 41 is significantly shifted from its native position and bound to a density of globular protein. This density can be reasonably fitted by an archaeal r-protein L1 (*Nikulin et al., 2003*) (*Figure 11G*, *Figure 1—figure supplement 5F*) and was assigned to Utp30 that contains a L1-like domain.

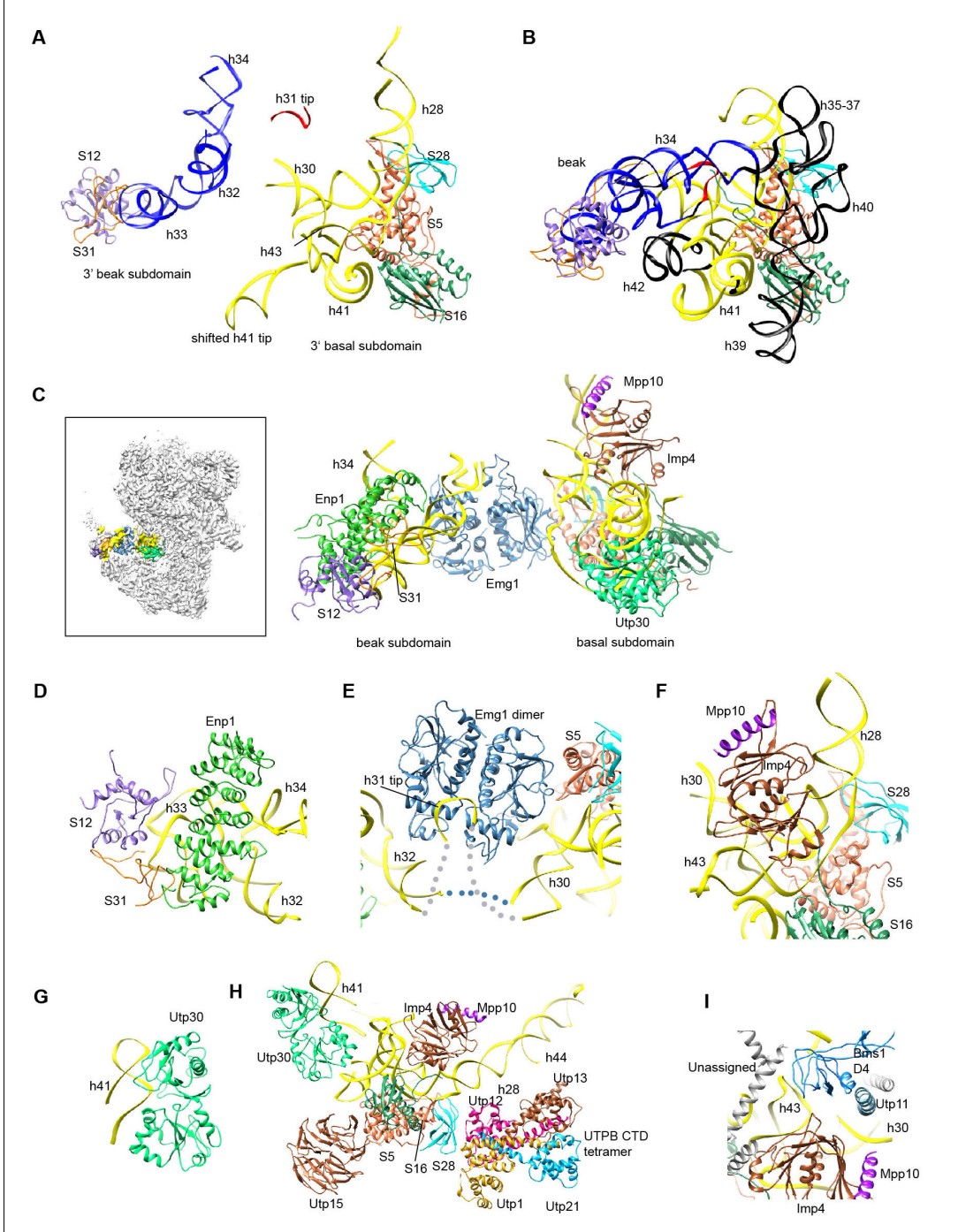

**Figure 11.** Structure of the 3' major domain in 90S. (**A**) The 3' major domain structure in 90S. RNA segments are color-coded. (**B**) The 3' major domain structure in 40S. Aligned to the 3' basal subdomain in A. The RNA regions undetected in 90S are colored black and other regions are colored as their counterparts in A. Only the 5 r-proteins detected in 90S are displayed. (**C**) The 3' major domain with closely associated AFs. The insert shows the displayed region in the ΔDhr1 map. (**D**) Interaction of Enp1 with the beak. (**E**) Interaction of Emg1 with the 3' major domain. (**F**) Interaction of Imp4 with the 3' basal subdomain. (**G**) Interaction of Utp30 with helix 41. (**H**) Interaction of the 3' basal subdomain with UTPA and UTPB components. (**I**) Interaction of the 3' basal subdomain with Bms1 and Utp11.

The 3' basal subdomain is placed at the front face by interactions with the 5' ETS factors and the late factor Bms1 (*Figures 4F* and *9D*). S5 docks on the WD domain of the UTPA protein Utp15 (*Figure 11H*). The CTD of the UTPB protein Utp12 contacts S28 and helix 28 (*Figure 11H*). A long helix of Utp11 binds at helix 30 (*Figure 11I*). The D4 domain of Bms1 inserts into a pocket composed of helix 43, Imp4 and Utp11 (*Figure 11I*).

## The 3' minor domain

The 3' minor domain consists of helices 44 and 45 that both mediate long-range interactions in the mature 40S structure. Helix 44, the longest one in 18S, joins the body, and helix 45 constitutes part of the platform. In 90S, helix 44 follows the direction of its neighboring helix 28 in the 3' major domain and travels in a long curved path to rest its tip on Utp22, forming the small arch(*Figure 3*, *Figure 12A*). The basal part of helix 44 is clamped between the tandem WD domains of Utp12 and Utp13 but primarily contacts the inner surface of the tandem WD domains of Utp12.

The KH domain protein Pno1 binds at the platform near the D site in the pre-40S particle (*Strunk et al., 2011*) and crosslinks with the D site sequence (*Turowski et al., 2014*). In 90S, Pno1 is located near the center of UTPB with interactions to the CTD of Utp21 and the WD domain of Utp1 (*Figure 12B*). A density that is suggestive of a short RNA hairpin is sandwiched between Pno1 and the WD domain of Utp13 (*Figure 1—figure supplement 5N*). Because the D site RNA potentially forms a short hairpin (*Figure 8—figure supplement 1*) and binds Pno1, the density was temporarily assigned to the D site RNA. Although the D site RNA was found by chemical probing to be single-stranded (*Lamanna and Karbstein, 2009*), this result may mainly reflect the structure of pre-rRNA in the more abundant pre-40S ribosome. In the 90S model, the first KH domain of Pno1, which contains a degenerated RNA-binding motif, contacts the duplex part of the hairpin. The second KH domain of Pno1 binds at the 5' single-stranded region of 18S rRNA with the conserved RNA-binding surface, consistent with the known RNA-binding model of KH domain. The linker between helix 44 and D site and major part of ITS1 were not detected. During the stepwise assembly process of 90S, Pno1 is bound after the transcription of helix 44 and before the emergence of the D site sequence (*Zhang et al., 2016b*). This suggests that Pno1 is initially recruited by the protein-protein interaction with the UTPB complex.

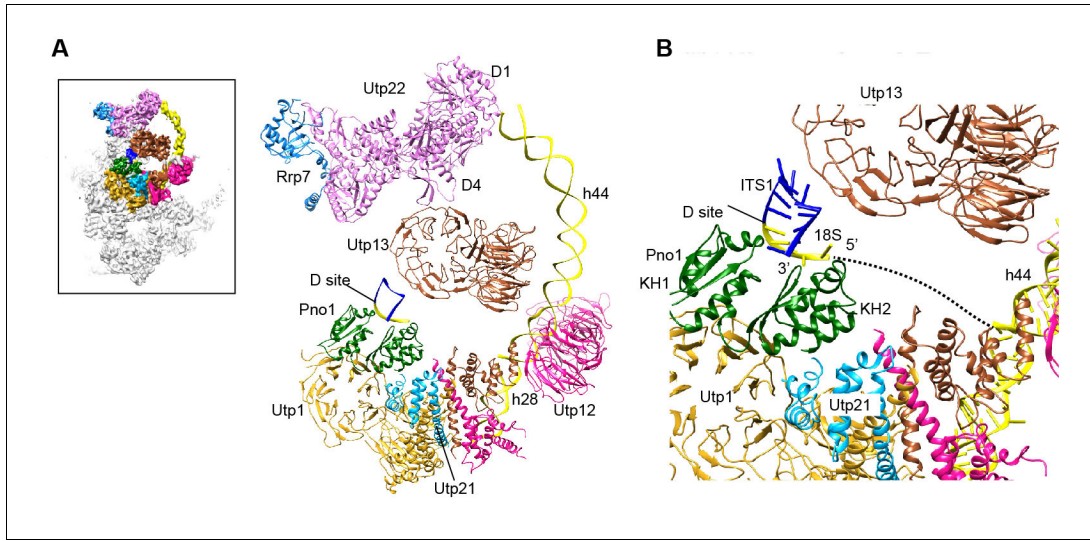

**Figure 12.** Structure of the 3' minor domain in 90S. (**A**) Assembly of helix 44 and the D site hairpin. The 18S and ITS1 RNA are colored yellow and blue. The insert shows the displayed region in the ΔDhr1 map. (**B**) Structure of Pno1 and the D site hairpin. The dotted line indicates a potential path of helix 45.

## Discussion

The 90S model contains a large fraction of stably associated AFs and most likely represents the complete 90S particle (*Zhang et al., 2016b*). One main function of 90S is to process the pre-rRNA transcript at the A0, A1 and A2 sites. Unfortunately, the current model does not include these processing sites that appear to locate at the periphery or outside of the 90S core structure. Although the three analyzed 90S samples contain multiple processing intermediates of pre-rRNA (*Figure 1F–G*), all reconstructions led to similar structures, suggesting that pre-rRNA cleavage may not significantly affect the global architecture of 90S.

The 90S model provides significant insight into the early assembly principle of small subunit. The four domains of 18S are assembled with 19 core r-proteins into isolated subdomain structures that correspond to the early folding units of small subunits. These structures are generally similar to the corresponding structure in the mature subunit, although deviations in local structure are prevalent. The 5' domain and the central domain are assembled to great extents and primed to combine into the body. In contrast, the 3' major domain and the minor domain are significantly deviated from their native structures. Large structural reorganization is required to establish the global architecture of 40S during the subsequent 90S-to-pre-40S transition.

The current 90S model includes structures for 36 AFs and the 5' ETS and U3 RNA, providing major insight into the function of these components in ribosome assembly. Many AFs directly bind to the nascent ribosome and appear to stabilize the partially assembled structures at stages when some contacts present in the mature subunit have not been established. For example, Krr1 appears to substitute the structural role of helix 45 and S26 in the partially assembled platform.

The 5' ETS subcomplex acts as a large base to support and arrange individual ribosomal substructures in 90S. U3 is a prominent component in the 5' ETS subcomplex (*Chaker-Margot et al., 2015*; *Zhang et al., 2016b*). The 5' domain of U3 is located at the heart of 90S and crucially organizes its architecture by making four helices with the 5' ETS and 18S RNA. We show that U3 binds 18S RNA in a way different from the previously proposed model (*Figure 6—figure supplement 1*). In the mature 40S structure, the 5' end of 18S base pairs with a central region (nt 1140–1144) to form the central pseudoknot (CPK). U3 has been long thought to bind both sides of the CPK and chaperone its formation (*Hughes, 1996*). However, our structure shows that U3 pairs with the 5' end of 18S, but not with the 3' side of the CPK. Previous mutational studies also support the former, but not the latter, interaction (*Sharma and Tollervey, 1999*). Unexpectedly, we found that the 5' end of U3 makes a novel evolutionarily conserved interaction with the helix 27 sequence upstream of the 3' side of the CPK. By bringing the 5' end and the central region of 18S close, U3 would facilitate the formation of the CPK and the overall architecture of mature 40S after cleavage of sites A0-A2 and during the transition of 90S to pre-40S.

The 90S structure assembles in a stepwise manner as the pre-rRNA transcript extends in 5' to 3' direction (*Chaker-Margot et al., 2015*; *Zhang et al., 2016b*). The structural organization of 90S is consistent with the ordered assembly pathway of the 5' ETS, the 5' domain and the central domain as these RNA sequences can form rather independent structures with the associated AFs (*Figure 13*). The transcription of helix 44 was shown to trigger a dramatic assembly event, which involves association of 12 late factors and release of U14, snR30 and about 14 AFs bound earlier, and eventually leads to the compaction of 90S particle and acquirement of cleavage competency (*Osheim et al., 2004*; *Zhang et al., 2016b*). The structural role of helix 44 and six assigned late factors (Utp30, Enp1, Bms1, Rcl1, Pno1, Utp20) in the 90S structure provide insight into the last major assembly event. They function to assemble the 3' major domain (Utp30, Enp1), the 3' minor domain (Pno1) and the 5' basal subdomain (Bms1) and to establish long range connections between ribosomal subdomains (helix 44, Bms1/Rcl1, Utp20). Specifically, helix 44 links helix 28 to Utp22, establishing a connection between the 3' major domain and the central domain. Binding of helix 44 to the tandem WD domains of Utp12 and Utp13 would lock their conformation. Consequently, Utp13 would promote the association of Pno1 and the D site RNA and Utp12 would induce the closure of the front face by stabilizing the Bms1/Rcl1 complex that also connects the 5' basal subdomain and the 3' basal subdomain. The elongated Utp20 molecule ties up the central domain and the 5' domain. These assembly events, triggered by helix 44, likely occur cooperatively to establish the global architecture of 90S.

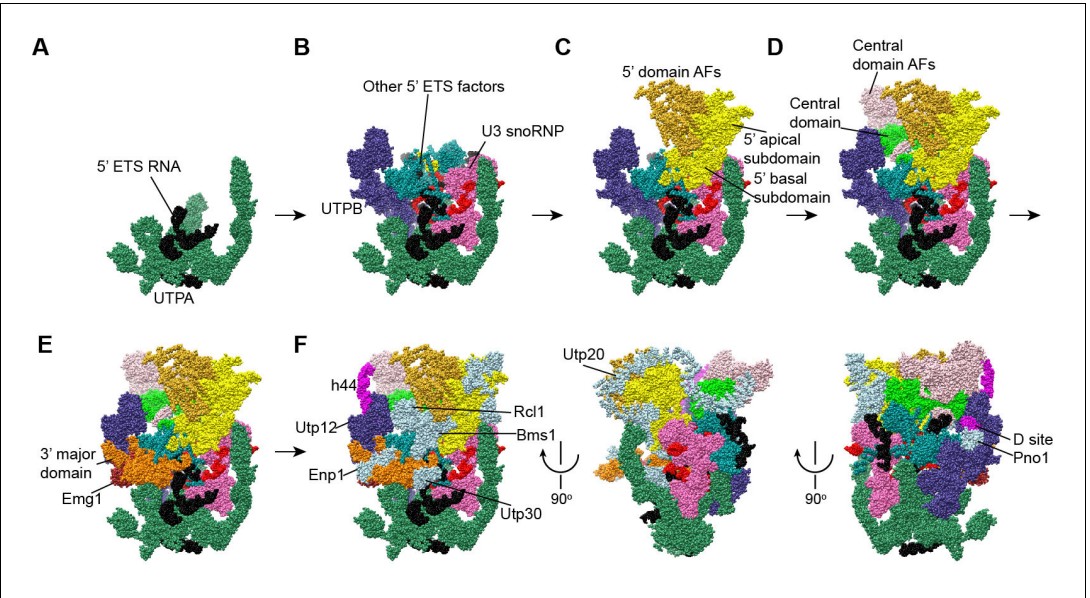

**Figure 13.** Stepwise assembly of 90S. (**A–F**) Components are added by their known assembly order. Assembly of UTPA (**A**), UTPB, U3 snoRNP and other 5' ETS factors (**B**), the 5' domain and its AFs (**C**), the central domain and its AFs (**D**), the 3' major domain and its AFs (**E**) and the 3' minor domain and the late factors (**F**). The view is rotated by −45° along y axis from the front view. The fully assembled 90S is additionally shown in two perpendicular views. Late AFs are labeled in **F**.

The 90S maps from Sc and Ct are largely similar (*Figure 1—figure supplement 6C*) (*Kornprobst et al., 2016*), indicating the conservation of 90S architecture in yeast. However, the Sc map is more complete than the Ct map. The Ct 90S map lacks densities for the Utp22-Rrp7 complex, Rrp5, Utp18 (in our assignment) and helix 44 and has extremely weak densities for the beak subdomain and bound Enp1. These differences in the density map may be caused by species-specific features, different maturation stages of purified 90S and different sample preparation procedures. The two models have different assignment for Utp6, Utp18, the N domain of Utp10 and Kre33. The bulky density around the 5' apical major domain was assigned to two Kre33 molecules in the Ct 90S, but was tentatively assigned to Bfr2, Lcp5 and Efg1 in our study. We have modeled the 5' ETS and the 5' domain of U3 snoRNA and assigned all UTPA proteins, Sof1, Utp7, Utp11, Enp2, Enp1, Utp22, Rrp7, Rrp5, Krr1 and Pno1. Our model is based on the cryo-EM maps of better resolution and completeness, additional high-resolution crystal structures and chemical crosslinking data and is of much higher quality over the Ct model. Nevertheless, some assignments, especially those without high-resolution crystal structures, should be considered tentative at the current resolution of cryo-EM map.

The nearly complete cryo-EM structure of 90S provides a platform to understand the large body of genetic and biochemical data on 18S rRNA processing and SSU assembly and sets the stage to investigate the mechanism and dynamics of early SSU assembly at the level of entire 90S structure.

## Materials and methods

### Yeast strains

All strains were derived from BY4741 (Mat a, leu2Δ0, Met15Δ0, ura3Δ0). The Noc4-TAP (BY4741, Noc4-TAP::HisMX) and Enp1-TAP (BY4741, Enp1-TAP::HisMX) strain were purchased from Open Biosystems. The conditional expression strains GAL::dhr1/Enp1-TAP (BY4741, Enp1-TAP::HisMX, natNT2-pGALL::3HA-dhr1) and GAL::mtr4/Enp1-TAP (BY4741, Enp1-TAP::HisMX, natNT2-pGALL::3HA-mtr4) were generated by one-step PCR strategy. A *GAL* promoter cassette containing the *natNT2* gene, the *GALL* promoter and a 3xHA-tag was PCR-amplified from plasmid pYM-N28

with gene-specific primers (*Janke et al., 2004*). The Enp1-TAP strain was transformed with the PCR product and selected for nourseothricin resistance. The genomic tag was confirmed by PCR and the HA tag on expressed proteins was checked by western blot.

## Purification of 90S particles

The Noc4-TAP cells (six liters) were grown at 30°C in YPD medium (1% yeast extract, 2% peptone, 0.003% adenine, 2% glucose) to $OD_{600}$ of 2.5–3.5. To deplete Dhr1 or Mtr4, the GAL::dhr1/Enp1-TAP or GAL::mtr4/Enp1-TAP cells were first cultured in YPG medium (1% yeast extract, 2% peptone, 0.003% adenine, 2% galactose). The cells were harvested, washed twice with sterile water, transferred into 6 liters of YPD and further cultured to $OD_{600}$ of 0.8–1.0 for 14 hr.

The cells were harvested and resuspended in lysis buffer (20 mM HEPES-K pH8.0, 110 mM KOAc, 40 mM NaCl) supplemented with one tablet of EDTA-free protease inhibitor cocktail (Roche). The cells were lysed by a high pressure cell disruptor (JNBIO) followed by centrifugation at 6,000g for 20 min. The resultant supernatant was incubated with 25 mg IgG-coated magnetic Beaver beads (Beaverbio) or Dynabeads (Invitrogen) for 30–45 min. The beads were washed with lysis buffer three times and incubated with TEV protease in cleavage buffer (20 mM HEPES-K, pH 8.0, 100 mM KOAc and 5 mM DTT) at 4°C overnight. The eluate was concentrated to about 50 µl.

The Noc4-TAP particle for cyro-EM analysis was additionally purified by density gradient fractionation. The eluate was applied to a 4 ml 15–45% glycerol gradient prepared in buffer (50 mM HEPES-Na, pH 8.0, 100 mM NaCl) and centrifuged at 22,500 rpm for 14 hr in a SW Ti 60 rotor (Beckman). Fractions of 200 µl were collected manually and measured for absorbance. The peak fractions were exchanged to cleavage buffer using ultrafiltration devices to remove glycerol and were concentrated to 20 µl.

## Northern blot analysis

Northern blot analysis was conducted as described (*Lin et al., 2013*). IgG-coated Beaver beads were incubated with 200 $OD_{260}$ units of clarified cell lysates for 40 min and washed seven times with lysis buffer. RNA was extracted from the beads using TRIzol reagent and resolved in 1.2% agarose-formaldehyde gels for large RNA analysis or in 8% polyacrylamide-8 M urea gels for small RNA analysis. RNA was transferred to Hybond N+ membranes (GE Healthcare) and hybridized to 5'-$^{32}$P-labeled DNA probes. The following probes were used for hybridization: U3 (5'-GGATTGCGGACCAAGC TAA-3'), D-A2 (5'-CGGTTTTAATTGTCCTA-3'), A0-A1 (5'-AAAGAAACCGAAATCTCTTT-3') and 5'-A0 (5'-GGAAATGCTCTCTGTTCAAAAAGCTTTTACACTCTTGACCAGCGCACTCC-3').

## Negative stain EM analysis

The Noc4-TAP sample was first analyzed with negative stain EM. Aliquots of 3–5 µl sample ($OD_{280}$ = 1.5–2.0) was incubated with glow-charged grids for 30 s and stained with 2% uranyl acetate. Grids were transferred to an FEI Tecnai T12 electron microscopy operated at 120 kV. Images were recorded on a 2k X 2k Gatan 894 CCD camera with a pixel size of 2.49 Å and defocus of 2–4 µm. 15,654 particles were picked from 507 images using e2boxer.py and corrected for contrast transfer function (CTF) in EMAN2 (*Tang et al., 2007*). Particles were subjected to reference-free 2D classification in RELION. After cleaning up, 14,037 particles were 3D classified into two classes in RELION using the yeast 40S ribosome structure low-pass filtered to 80 Å as the initial model (*Ben-Shem et al., 2011*). One class with higher resolution displayed reasonable features and was refined to 25 Å in RELION (*Figure 1—figure supplement 1*).

## Cryo-EM data collection

To prepare vitrified specimen, aliquots of 3 µl sample ($OD_{280}$ = 1.5–2.0) were incubated for 30 s on glow-charged holey carbon grids (Quantifoil 2/2 or Quantifoil 1.2/1.3) covered with a home-made continuous thin layer of carbon (~5 nm thick) at 4°C and 100% humidity. Grids were blotted for 2 s and rapidly plunged into liquid ethane using an FEI Vitrobot Mark IV.

The cryogenic Noc4 sample was first screened on a Talos F200C 200-kV electron microscope equipped with a 4 K × 4 K Ceta camera (FEI). Several datasets were collected with a pixel size of 2.27 Å and defocus values of 2–4.5 µm. The particles were subjected to 3D classification in RELION using the 40S ribosome structure as the initial model. Finally, 8910 particles were selected to

reconstruct a density map at 13.3 Å overall resolution. The WD domains can be readily recognized in this map, validating the choice of the initial model and the correctness of the map. The map was filtered to 60 Å and used as an initial model for subsequent reconstructions.

High resolution data were collected on an FEI Titan Krios electron microscope operated at 300 kV. Images were automatically acquired with SerialEM in the low-dose mode (*Mastronarde, 2005*). The images for the Noc4-TAP sample were recorded on a Falcon II direct electron detector at a nominal magnification of 52,000, corresponding to a pixel size of 1.42 Å. Defocus ranged from 1.5 to 5 μm. Images were exposed for 1.6 s with a total electron dose of 40 e/Å$^2$. 1769 images were collected and 127,198 particles were used for 3D classification.

The ΔDhr1/Enp1-TAP data were collected at a nominal magnification of 45,000 on a Falcon III direct electron detector with a pixel size of 1.76 Å and defocus of 2.0 to 4.0 μm. Images were exposed for 2 s with a total electron dose of 50 e/Å$^2$. 2055 images were collected and 420,755 particles were used for 3D classification.

The ΔMtr4/Enp1-TAP data were collected at a nominal magnification of 45,000 on a Falcon III direct electron detector with a pixel size of 1.76 Å and defocus of 1.5 to 3.5 μm. Images were exposed for 1.6 s with total electron dose of 40 e/Å$^2$. 1102 images were collected and 195,317 particles were used for 3D classification.

## Cryo-EM image processing

Raw image stacks were aligned for motion correction at micrograph level using MOTIONCORR (*Li et al., 2013*). The CTF parameters of motion-corrected images were measured by CTFFIND3 or CTFFIND4 (*Mindell and Grigorieff, 2003*). Particle autopicking, 2D and 3D classification, 3D refinement and postprocess were conducted with RELION (*Scheres, 2012a*; ).

After CTF parameter measurement, images with poor resolution were discarded. Particles were autopicked in RELION. The templates for autopicking were obtained from 2D class averages of 1000–2000 manually picked particles. After autopicking, micrographs were inspected to remove bad particles and contaminations. Particles were extracted with a box size of 480 pixels for the Noc4-TAP dataset and a box size of 400 pixels for the ΔDhr1 and ΔMtr4 datasets. Particles were downsized by 2–4 folds for 2D and 3D classification. 3D classification was performed for 1–3 rounds.

Particles that belong to the classes of high resolution and similar structures were selected for the next around of classification or refinement. The final reconstruction was refined with the gold-standard FSC method (*Scheres and Chen, 2012*).

The refined map was subjected to the postprocess procedure in RELION to correct the modulation transfer function of detector and to sharpen B-factor. B-factor was calculated automatically according to the Guiner Plot above 10 Å. A soft-edge mask with 3-pixel extension and 5-pixel falloff was applied for resolution evaluation. The reported resolution was based on the gold-standard FSC = 0.143 criterion. Local resolution was estimated with ResMap (*Kucukelbir et al., 2014*).

12,643 particles of the Noc4-TAP dataset were used to reconstruct a 8.7 Å density map (*Figure 1—figure supplement 2*, *Supplementary file 3*). 30,995, 49,514 and 31,700 particles of the ΔDhr1 dataset were used to reconstruct three maps (states 1, 2 and 3) at 7.8, 8.7 and 9.5 Å, respectively (*Figure 1—figure supplement 3*). 73,543 particles of the ΔMtr4 dataset were refined to yield a 4.5 Å density map (*Figure 1—figure supplement 4*).

## Model building

The tetrameric structure of the CTD of Ct Utp1, Utp21, Utp12 and Utp13 of UTPB was fit into the EM density (*Zhang et al., 2016a*). Four tandem WD domains were found around the CTD tetramer. The crystal structure of yeast Utp21 tandem WD domain was fitted as rigid body (*Zhang et al., 2014*). Based on the CXMS data, the tandem WD domain that contacts Utp21 was assigned to Utp1 and a single WD domain contacting Utp21 was assigned to Utp18. The tandem WD domain of Utp12 was linked to its CTD by visible densities and Utp13 WD domain was subsequently assigned. The HAT domain protein Utp6 was not located (*Zhang et al., 2016a*).

A helical structure was de novo built for the CTD tetramer of UTPA. Two tandem WD domains and three WD domains were identified at the jaw region. The WD domains of Utp15, Utp5 and Utp8 and the tandem WD domain of Utp4 were assigned by the CXMS data (*Supplementary file 1*). The other tandem WD domain was assigned to Utp17. The CTDs of Utp5, Utp15 and Utp8 were

assigned by spatial proximity to their WD domains and protein interaction relationship (*Pöll et al., 2014*). Utp9 appears to miss a WD domain. The crystal structures of the N domain of ctUtp10 bound with a C-terminal peptide of ctUtp17 and the M domain of ctUtp10 were fitted in the density map. The C domain of Utp10 was modeled as two copies of the superhelix structure of importin Kap60 (*Matsuura and Stewart, 2005*).

U3 snoRNP is modeled by an archaeal C/D RNP structure (*Lin et al., 2011*). Three subcomplexes of the C/D RNP structure: the coiled-coil dimer, the Nop5 NTD/fibrillarin complex, the Nop5 CTD/ L7Ae/K-turn RNA complex, were separately docked as a rigid body. The WD domain structure of yeast Rrp9 was fitted with unambiguous orientation (*Zhang et al., 2013*). P1, P2 and the basal parts of P4 and P5 of U3 RNA were modeled as A-form RNA duplex. P3 was not modeled. Nop56 and Nop58 were assigned by their asymmetric binding to box B/C and box C'/D (*Cahill et al., 2002*).

The structures of other AFs or their homologous proteins were first docked as rigid body in Chimera (*Supplementary file 1*). Emg1, S5 and S16 were initially located with the BALBES-MOLREP pipeline (*Brown et al., 2015*). If a homologous structure is available, the structure of Sc proteins was modeled with SWISS-MODEL or MODELLER (*Supplementary file 1*) (*Eswar et al., 2007*; *Biasini et al., 2014*). Helical structures were de novo built for the CTD complex of UTPA, Utp20, the N-terminal region of Utp11 and other unassigned densities (Unk1, Unk2 and others). A-form RNA helices were built for 5' ETS and U3 RNA. To improve the 5' ETS RNA structure, each 5' ETS helix was predicted for secondary structure with RNAfold (*Gruber et al., 2008*) and folded into 3D structure with RNAComposer (*Popenda et al., 2012*). To model the ribosome structure, substructures were cut out from the yeast 40S structure (*Ben-Shem et al., 2011*) and docked into the density in Chimera. The structural elements with no or very weak density were removed.

Model building was conducted in Coot (*Emsley and Cowtan, 2004*). The model was fitted to the density with molecular dynamic flexible fitting (MDFF) (*Trabuco et al., 2008*) and finally refined against the density map by using phenix.real_space_refine with secondary structure and geometry restraints. The high-resolution ΔMtr4 map was primarily employed for refinement. The ΔDhr1 (state 1) map was used for the regions with weak density in the ΔMtr4 map. Structural figures were prepared with Chimera and PyMOL (*DeLano, 2002*; *Pettersen et al., 2004*).

## Crosslinking of 90S

The CXMS experiment was performed as described (*Wu et al., 2016*). The ΔMtr4 90S sample containing ~10 µg total proteins was crosslinked with $BS^3$ or DSS. The cross-linked proteins were precipitated with acetone and subjected to tryptic digestion. LC-MS/MS analyses were carried out on an EASY-nLC 1000 system interfaced to a Q-Exactive HF mass spectrometer (Thermo Fisher Scientific, Waltham, MA). Cross-linked peptides were identified with the pLink software (*Yang et al., 2012*).

## Database

The cryo-EM density maps and PDB files of 90S have been deposited under accession numbers EMD-6695, EMD-6696, EMD-6697, 5WYJ and 5WYK.

## Acknowledgements

We would like to thank the Center for Biological Imaging (CBI), Institute of Biophysics, Chinese Academy of Science (CAS) for cryo-EM study and HPC-Service Station in CBI for image processing. We are grateful to Fei Sun, Xiaojun Huang, Zhenxi Guo, Gang Ji, Bolin Zhu, Shuoguo Li and Deyin Fan at CBI, Wanzhong He at NIBS, Ning Gao at Tsinghua University for help in EM sample preparation and data collection, Hongjie Zhang for help in radioactive experiments, Yan Guo for technical assistance and Niu Huang for sharing computation resources. This work was supported by the National Natural Science Foundation of China [91540201, 31430024, 31325007 to KY], the Strategic Priority Research Program of the Chinese Academy of Sciences [XDB08010203 to KY], the 100 Talents Program of CAS [to KY], the Ministry of Science and Technology of China [973 grant 2014CB84980001 to M-QD] and the Beijing Municipal Government.

# Additional information

## Funding

| Funder | Grant reference number | Author |
|---|---|---|
| Ministry of Science and Technology of China | 2014CB84980001 | Meng-Qiu Dong |
| National Natural Science Foundation of China | 91540201 | Keqiong Ye |
| Chinese Academy of Sciences | XDB08010203 | Keqiong Ye |
| National Natural Science Foundation of China | 31430024 | Keqiong Ye |
| National Natural Science Foundation of China | 31325007 | Keqiong Ye |

The funders had no role in study design, data collection and interpretation, or the decision to submit the work for publication.

## Author contributions

QS, Prepared samples, collected all EM data and constructed the ΔDhr1 and ΔMtr4 maps; XZ, Built and refined the model; JQ, WA, Prepared samples and conducted northern blot analysis; PL, Prepared samples; DT, Preformed CXMS analysis; RC, BW, SZ, CZ, XC, WZ, Conducted crystallographic studies; JC, Analyzed the secondary structure of 5' ETS; M-QD, Supervision, Preformed CXMS analysis; KY, Conceptualization, Supervision, Funding acquisition, Writing—original draft, Writing—review and editing, Constructed the Noc4 map, built the model

## Author ORCIDs

Xing Zhu, http://orcid.org/0000-0003-4839-4857
Meng-Qiu Dong, http://orcid.org/0000-0002-6094-1182
Keqiong Ye, http://orcid.org/0000-0001-6169-7049

# Additional files

## Supplementary files

• Supplementary file 1. Components and modeling of yeast 90S structure.

• Supplementary file 2. Chemical crosslinking and mass spectrometry data for the ΔMtr4/Enp1-TAP sample. Sheet one stores the crosslinked peptides. Sheet two shows the annotated intermolecular crosslinks. Sheet three shows the intramolecular crosslinks. The structural consistency for crosslinks between two assigned proteins is judged by the distance (<30 Å) between crosslinked sites. If a crosslinked site is unmodeled or undetermined in structure, the position of neighboring residues or domains is considered.

• Supplementary file 3. Statistics of data collection, structural refinement and model validation.

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
