## [Decision Letter]

Thank you for submitting your article "Molecular architecture of the 90S small subunit pre-ribosome" for consideration by *eLife*. Your article has been reviewed by four peer reviewers, one of whom, Alan Hinnebusch (Reviewer #1) and the evaluation has been overseen by James Manley as the Senior Editor. The following individuals involved in review of your submission have agreed to reveal their identity: David Tollervey (Reviewer #3) and Yaser Hashem (Reviewer #4).

The reviewers have discussed the reviews with one another and the Reviewing Editor has drafted this decision to help you prepare a revised submission.

Summary:

This paper describes a cryoEM reconstruction of the *S. cerevisiae* 90S pre-ribosome that was affinity purified using Noc4-TAP or Enp1-Tap as bait proteins, and in the latter case isolated from strains lacking processing factors Dhr1 or Mtr4 in an effort to stall the maturation at a discrete stage. Three similar maps were obtained and discussed here, a Noc4-TAP complex at 8.7A, a ΔDhr1 complex at 8.7A, and a ΔMtr1 complex at 4.5A. (Two other ΔDhr1 complexes were analyzed but are being presented elsewhere.) The ΔMtr1 complex was also analyzed by chemical crosslinking and MS (CXMS) to obtain 340 intermolecular and 255 intramolecular linkages to assist in assignment of EM densities. Crystal structures of 19 of the 33 40S ribosomal proteins (RPs) that could be identified in the structure, and of ~27 different assembly factors (AFs), including a number of unpublished x-ray structures determined by the authors and not yet deposited in RCSB, were modeled into the EM densities. Other densities corresponding to β-propellers were fitted with known structures of other WD40 repeat proteins, or were modeled as poly-alanine helical segments or A-form RNA helices. The structure contains a total of 19 RPs, 35AFs, the 5' external transcribed spacer (ETS), and the U3 RNA. A cryoEM model of *Chaetomonium thermophilum* (Ct) 90S at 7.3A was recently reported. It is stated that the current Sc map is more complete than the Ct map, containing densities not assigned in the Ct map. In addition, a number of densities are assigned differently here compared to the Ct map. Hence, the current map seems to represent a valuable addition to the literature of the 90S pre-ribosome.

Among the interesting findings is that the 4 subdomains of the mature 40S are evident in the 90S, with deviations in local structure of different extents for the different subdomains; but large-scale reorganization is required to achieve the interdomain arrangements of the mature 40S. They present evidence that U3 binds 18S rRNA in a manner distinct from a previous model and also does not interact with both sides of the central pseudoknot. The structure shows how the 90S can assemble co-transcriptionally in a step-wise fashion as the pre-rRNA transcript is produced. It is observed that helix 44 (h44), not visible in the Ct map, assembles with late AFs and establishes a long-range connection between the 3' major and central domains to help establish the global architecture of the 90S; and that h44 undergoes a major re-location between the 90S and mature 40S.

Essential revisions:

There was general agreement among the reviewers that the paper is potentially suitable for publication but that additional source data must be supplied to the reviewers to allow an evaluation of certain key assignments of electron density, that extensive revisions of text and figures will be required, and that Northern analysis of the rRNAs present in each of the solved complexes is needed. It is necessary for the authors to make the density map available to the reviewers to allow first-hand evaluation of some of the most unexpected assignments, including a revised base-pairing scheme for U3, a dramatic reorganization of helix 44, and a stem-loop at the D cleavage site. It is also imperative that the pdb files for all of the novel crystal structures be deposited prior to publication and be made available to the referees for inspection during the review of the revised manuscript. The authors also need to document the cross-linking/MS data and indicate when it was used to guide an assignment, and whether each crosslink is consistent with the final structure. The authors must do a better job of citing the previous literature. The paper is a challenge to read and each of the reviewers has specific suggestions for how the presentation of data can be improved, both in text and figures. In addition it is important that the authors address in a more thorough and transparent way the discrepancies between their assignments and those of the recent structure published by Kornprobst et al., and their objective justifications for their assignments. In short, the readers/reviewers of this paper must be able to assess the quality of the interpretation and the model from the data provided by the manuscript, knowing that some of these assignments are provisional, and the authors must do their best to facilitate this assessment. The essential revisions can be found in the reviews below.

Reviewer #1:

This paper describes a cryoEM reconstruction of the *S. cerevisiae* 90S pre-ribosome that was affinity purified using Noc4-TAP or Enp1-Tap as bait proteins, and in the latter case isolated from strains lacking processing factors Dhr1 or Mtr4 in an effort to stall the maturation at a discrete stage. Three similar maps were obtained and discussed here, a Noc4-TAP complex at 8.7A, a ΔDhr1 complex at 8.7A, and a ΔMtr1 complex at 4.5A. (Two other ΔDhr1 complexes were analyzed but are being presented elsewhere.) The ΔMtr1 complex was also analyzed by chemical crosslinking and MS (CXMS) to obtain 340 intermolecular and 255 intramolecular linkages to assist in assignment of EM densities. Crystal structures of 19 of the 33 40S ribosomal proteins (RPs) that could be identified in the structure, and of ~27 different assembly factors (AFs), including a number of unpublished x-ray structures determined by the authors and not yet deposited in RCSB, were modeled into the EM densities. Other densities corresponding to β-propellers were fitted with known structures of other WD40 repeat proteins, or were modeled as poly-alanine helical segments or A-form RNA helices. The structure contains a total of 19 RPs, 35AFs, the 5' external transcribed spacer (ETS), and the U3 RNA. A cryoEM model of *Chaetomonium thermophilum* (Ct) 90S at 7.3A was recently reported. It is stated that the current Sc map is more complete than the Ct map, containing densities not assigned in the Ct map. In addition, a number of densities are assigned differently here compared to the Ct map. Hence, the current map seems to represent a valuable addition to the literature of the 90S pre-ribosome.

Among the interesting findings is that the 4 subdomains of the mature 40S are evident in the 90S, with deviations in local structure of different extents for the different subdomains; but large-scale reorganization is required to achieve the interdomain arrangements of the mature 40S. They present evidence that U3 binds 18S rRNA in a manner distinct from a previous model and also does not interact with both sides of the central pseudoknot. The structure shows how the 90S can assemble co-transcriptionally in a step-wise fashion as the pre-rRNA transcript is produced. It is observed that helix 44 (h44), not visible in the Ct map, assembles with late AFs and establishes a long-range connection between the 3' major and central domains to help establish the global architecture of the 90S; and that h44 undergoes a major re-location between the 90S and mature 40S.

General critique:

The paper presents a large amount of structural information, of which a significant amount was apparently not obtained in the previous lower resolution structure of the Ct 90S and, hence, should be of considerable value to the field. The details of the model building are fairly well described in the Materials and methods section and [Supplementary-material SD1-data], with the exception that the manner in which the CXMS data was enlisted to justify the assignment of ambiguous densities was not documented or explained at all. This information could be added to [Supplementary-material SD1-data] to indicate which crosslinks tabulated in the Supplementary Dataset 1 were used to justify each assignment. Similarly, the Supplementary Dataset 1 should be annotated to indicate whether each crosslink is consistent with the final structure and if it was used to make any assignments of density. [Supplementary-material SD1-data] could also be expanded to include published interactions of each AF that were taken into account in density assignments. At a maximal resolution of 4.5A, it can be difficult to trace the proteins and so it is imperative that the authors thoroughly document everything that went into each assignment so that at least readers can judge how convincing each assignment is.

It also seems important that the pdb file be deposited in RSCB for every novel crystal structure being cited here and employed in density assignments.

The paper is a very difficult read. This is partly because the structure is so massive and there are so many noteworthy features to be discussed. However, there are many instances where the presentation could be improved.

Reviewer #2:

In this manuscript the authors use 3D cryo-EM to reconstruct a structure for a pre-40S subunit isolated from yeast lacking the TRAMP component Mtr4. The manuscript presents an exciting structure and makes several bold, and provocative claims.

I have several comments and concerns, some simply aimed at a better presentation of the data, others more substantial. I believe these changes will improve the impact of the manuscript, especially given that a similar structure has already been published.

1) I believe the FSC curves give an overestimate for the resolution of the dMtr4 complex, due to anisotropy in the sample, which is reflected in the bump at the bottom. This is of course also reflected in the so map in Figure 1. The authors should state a resolution range for each of their maps, as well as an average resolution. In addition they should show the res map for both the front and the back side of the molecules in Figure 1.

2) This comment leads to the most important one: The manuscript makes very provocative claims, most importantly about the base pairing of U3. It is impossible to evaluate this claim without having access to the map. This is a general comment for EM papers. Reviewers need to have access to the data, because they otherwise cannot evaluate them properly (we get access to biochemical data, sequencing data, why not structural data?). E.g. in the Kornprobst paper the authors incorrectly assigned Kre33. This is obvious once you look into the density (as the density is not symmetrical), but not necessarily from the manuscript. In any case, without access to the map it is impossible to judge whether the authors correctly assigned the helices base pairing with U3.

3) Along the same lines, the authors make a number of surprising assignments, including that of H44 at a very different location. This would be a neat finding, but again is hard to judge without a map. Even more surprising is the assignment of the D-site. I am sure the map provides confidence in a stem-loop. However, there are hundreds of stemloops in this molecule. How the authors decide it is a stemloop at the D-site, which has been essentially disproven to exist is totally unclear.

4) What is the RNA in the molecule? The authors should show a Northern blot.

5) The authors use a lot of "unpublished crystal structures" to fit their structure. These structures need to be shown as supplemental data, and pdb files need to be submitted and referenced herein. Examples are: the Utp10 structure, Enp1 structure, the Tsr1 structure, the RRp5-TPR structure, the Imp3 and Rpf1/Rrs1 structure.

6) Subsection “U3 snoRNP”: "The maps were essentially superimposable." This claim needs a Figure.

7) Imp3 and Imp4 are in two different Figures and it is unclear how they relate to each other. Correll and co-workers have shown that they function together. Thus, they should be shown together. There should also be references to that work. As well as previous work from the Baserga lab, which predicted U3 and 18S interactions with Imp3 and Imp4.

Presentation issues:

1) The overview Figure 2 is presented as done by Kornprobst. This Figure, in my mind was presented really poorly, and I would strongly urge the authors to give a different view. I realize that this will be effort as they need to remake some Figures, but I think it will lead to most people reading this paper and not the Kornprobst paper. Specifically, I would like to suggest that they orient the complex such that the 5'-domain is in the canonical view from the subunit interface (as in Figure 8), where it is at the bottom and not at 2 o'clock (on an imaginary clock). This would also require a rewrite of the overall structure description. I really think it will be worthwhile as it will help people to think about the molecule.

2) Of course this would also require different view for Figure 3, Figure 4, Figure 8.

3) In addition, I would strongly urge the authors to include a panel in most of the other Figures to help the reader orient him/herself with respect to that view (Figure 4, Figure 5, Figure 7, Figure 9–Figure 11. This would be an overview Figure, its turn to achieve the presented view, and the zoombox in that turned view.

Reviewer #3:

The authors report an impressive structural analysis of early pre-ribosome structure, combining cryo-EM and CXMS. The work provides numerous insights and will doubtless inform many future studies. I have only minor suggestions for improvement.

Specific points:

1) Noc4-TAP was purified from cells at OD600 = 2.5-3.5 and Dhr1 and Mtr4 depletions from cells 0.8-1.0. Ribosome biogenesis, especially co-transcriptional processing, is known to depend on growth phase. Do the authors see any important differences between Noc4-TAP and Enp1-TAP Dhr1 depl (or Mtr4 depl) models that might reflect the preponderance of cotranscriptional vs. posttranscriptional cleavage?

2) Subsection “Structure of 5' ETS and U3 RNA”: H5, H10 and its flanking sequences to the A0 and A1 sites were not located. Is it possible that Enp1-TAP Dhr1 depleted particles are dead-end intermediates in which A0 and A1 cleavage has already occurred? Northern blot analysis of the RNA content of these particles would have been useful.

3) Figure 8—figure supplement 1: Structure probing previously showed that there is no secondary structure of RNA at site D.

4) Could the authors discuss Pno1 binding to D site shown on Figure 1—figure supplement 5 in relation to the finding from Zhang et al. (2016)? Also, Pno1 binding is in agreement with previous cross-linking experiments.

5) The potential roles of Utp24 and Rcl1 in pre-rRNA cleavage has recently been debated. Although the structure lacks the actual A1 and A2 cleavage sites, can the authors usefully comment on the positions of Rcl1-Bms1 and Utp24 relative to these sites?

*Reviewer #4:*

The manuscript by Ye et al. Reports the structure of the 90S complex, an early maturing pre-40S complex, from *S. cerevisiae* by cryo-EM at an average resolution of 4.5Å. In their work, Ye et al. reveals an intermediate to high -resolution structure of this fascinating complex composed of numerous proteins and RNA where a partially folded 40S structure can be observed engulfed by a very sophisticated scaffold made by the multiple assembly factors (AFs). The authors have attempted several affinity purifications and were successful in pulling down a homogeneous complex, depleted in Mtr4 that appears to yield similar complexes compared to their two other δ-Dhr1 and Noc4-TAP alternative purifications.

Overall, the paper is very rich in content and presents a highly interesting structure that was thoroughly analyzed. This work follows up on the recent the work by Kornprobst et al. published in Cell last July 2016 from the Beckmann and Hurt labs. The structure by Ye et al. although submitted several months after the Kornprobst et al. presents significant differences that in my opinion warrant its publication in *eLife*. One can only regret that these two studies were not submitted as back-to-back publications in the same journal.

The structure by Ye et al. presents an average overall resolution of 4.5Å, which is almost 3Å better than the structure by Kornprobst et al. derived from *Chaetomium thermophilum*. The structure by Ye et al. is "more complete", as several additional densities can be observed compared to the structure reported by Kornprobst et al. Namely, the densities attributed to Utp22 and h44 are unresolved in the latter study. Also, the two models (Yet et al. and Kornprobst et al.) have "different assignment" for Utp6, Utp18, the N-domain of Utp10, Utp24 and Kre33 ("The bulky density around the 5' apical major domain was assigned to two Kre33 molecules in the *C. thermophilum* 90S, but was tentatively assigned to Bfr2, Lcp5 and Efg1 in our study").

The interpretation of this so-far elusive complex remains to some extent arbitrary, similarly to the study by Kornprobst et al. Indeed, it's not clear how the authors can be sure of their assignment of the rRNA and some proteins and AFs. However, the assignment was made according the same standards compared to those of Kornprobst et al., and it's really difficult to judge of the accuracy of the assignment in both cases. Both studies present various important discrepancies, and the interpretation was mainly based on shape similarity to solved structures of ribosomal proteins and AFs, which can be accurate provided that the resolution of the map is sufficient. In the case of Ye et al., the resolution is higher than in the study by Kornprobst et al., which may suggest a more accurate assignment. However, some features were interpreted in both papers based on structural similarities to "predicted models", which is risky and might be revealed later as inaccurate. The presence of cross-linking data validates some of these interpretations, however cross-linking can be inaccurate as observed in numerous cases for several reasons.

Other than the contrasted assignments, Ye et al. were able to interpret more in depth the different constitutive complexes of the 90S. Indeed, UTP-B is interpreted in more details, such as the CTD tetramer of UTP-B composed of Utp 21, 13, 1 and 12, a feature that was only referred to as "UTP-B oligo" in Kornprobst et al. Also Ye et al. have modeled the 5' ETS and the 5' domain of U3 snoRNA and assigned all UTP-A proteins, Sof1, Utp7, Utp11, Enp2, Enp1, Utp22, Rrp7, Rrp5, Krr1 and Pno1.

The interpretation of the U3 snoRNP is consistent between both papers.

Briefly, I would like to pledge in favor of the manuscript by Ye et al. for the good reason that the community accepted the assignment method reported by Kornprobst et al., and it would only be fair to accord the same credit to Ye et al., but one must keep in mind that some assignments might be inaccurate and have to be handled with caution. The reported assignment must be considered as a first basis of work that can later be revealed accurate or inaccurate, and these two papers must be considered together. In future studies, it will be necessary to perform more direct experiments to validate some of the uncertain assignments, such as immuno-cryo-EM.

I only have one major concern:

Ye et al. argue that some of the assignment differences relatively to Kornprobst et al. could be due to species-specificity although the 90S appears very well conserved. While politically correct, I find this statement a bit weak and invite Ye et al. to provide arguments and to discuss the chosen contrasted assignments compared to those by Kornprobst et al. Indeed it is capital that the reader understands their assignment and whether Ye et al. believes that their interpretation is more accurate and why! Otherwise the assignment appears only arbitrary and a matter of personal preference.

[Editors' note: further revisions were requested prior to acceptance, as described below.]

Thank you for resubmitting your work entitled "Molecular architecture of the 90S small subunit pre-ribosome" for further consideration at *eLife*. Your revised article has been favorably evaluated by James Manley (Senior editor), a Reviewing editor, and one reviewer.

The manuscript has been substantially improved but there are few remaining issues raised by reviewer #2 that need to be addressed before acceptance, as outlined below:

1) (Relating to the following comment in the previous review) "The manuscript makes very provocative claims, most importantly about the base pairing of U3. It is impossible to evaluate this claim without having access to the map." My inspection of the map indicates that it does not support the claim that U3 base pairs with H27. Rather, the map shows a duplex, which the authors have assigned as U3 and H27. However, there is no way of telling that H27 is part of this duplex, because there is no density connecting it to either H26 or H28, and indeed 20 nt are missing on either side. In addition, it is unclear that U3 is the other part of the duplex, because there is no EM density for parts of it. The authors must remove the claim that U3 base pairs to helix 27. Similarly, the map does not support the placement of Mpp10, as it is free floating, and the authors should make the speculative nature of their assignment of Mpp10 helices more clear. As in: "Based on the known interactions with Imp3 and Imp4, we assign these to Mpp10, although they could come from other nearby proteins as well."

2) (Relating to the following comment in the previous review) "The authors make a number of surprising assignments, including that of H44 at a very different location." In their response to this comment the authors indicate that H44 has been assigned based on its interaction with Pno1, which is assigned confidently. However, Pno1 has been shown to bind single stranded RNA, not double stranded RNA, and Lamanna and Karbstein have shown that D-site is single stranded not in a hairpin. Thus, there is no "predicted" duplex. Reviewer 3 had similar concerns. Moreover, looking at the map, I actually have no faith that this is a hairpin. There is density extending from the loop, which shouldn't be there, if it really were a hairpin. I strongly suggest the authors take out the assignment of the D-site hairpin. Minimally, the authors must present this logic in the paper, and add that the identified density might be a hairpin, and if so, could be the D-site hairpin, although previous data on later intermediates indicate that at that stage D-site is single stranded. In other words, this interpretation rests on assumptions, which are unproven, and probably untrue.

3) (Relating to the following comment in the previous review) "What is the RNA in the molecule? The authors should show a Northern blot." Interestingly, even though the majority of the RNA is 22S, the model still has the 5'-ETS. Thus, either it remains bound through non-covalent interactions, or the EM preferentially shows a minor population. The authors must discuss this.

4) Discussion section: Osheim did not show that transcription of H44 brings by a compaction event; Osheim showed that there is compaction after transcription of nearly all of ITS1, and that cleavage requires transcription of 25S.

---

## [Author Response]

*Essential revisions:*

*There was general agreement among the reviewers that the paper is potentially suitable for publication but that additional source data must be supplied to the reviewers to allow an evaluation of certain key assignments of electron density, that extensive revisions of text and figures will be required, and that Northern analysis of the rRNAs present in each of the solved complexes is needed. It is necessary for the authors to make the density map available to the reviewers to allow first-hand evaluation of some of the most unexpected assignments, including a revised base-pairing scheme for U3, a dramatic reorganization of helix 44, and a stem-loop at the D cleavage site. It is also imperative that the pdb files for all of the novel crystal structures be deposited prior to publication and be made available to the referees for inspection during the review of the revised manuscript. The authors also need to document the cross-linking/MS data and indicate when it was used to guide an assignment, and whether each crosslink is consistent with the final structure. The authors must do a better job of citing the previous literature. The paper is a challenge to read and each of the reviewers has specific suggestions for how the presentation of data can be improved, both in text and figures. In addition it is important that the authors address in a more thorough and transparent way the discrepancies between their assignments and those of the recent structure published by Kornprobst et al., and their objective justifications for their assignments. In short, the readers/reviewers of this paper must be able to assess the quality of the interpretation and the model from the data provided by the manuscript, knowing that some of these assignments are provisional, and the authors must do their best to facilitate this assessment. The essential revisions can be found in the reviews below.*

1) We have provided northern blot analysis of RNAs present in the three 90S particles (Figure 1) and discussed the results.

2) We have provided the pdb file of 90S and cryo-EM maps for assessment.

3) All novel pdb files have been deposited.

4) The crosslinking/MS data has been annotated. The assigned proteins in [Supplementary-material SD1-data] are supplied with key crosslinking data.

5) The texts and figures have been revised according to reviewers’ comments. More references are cited.

6) Comments on the quality of our model were added at the end of Discussion.

7) The quality of the 90S model has been further improved. Homology structures were built for many yeast proteins ([Supplementary-material SD1-data]) and the RNA model was more complete. The 90S structure has been refined against the density map by MDFF and Phenix ([Supplementary-material SD3-data]).

*Reviewer #1:*

*[…] General critique:*

*The paper presents a large amount of structural information, of which a significant amount was apparently not obtained in the previous lower resolution structure of the Ct 90S and, hence, should be of considerable value to the field. The details of the model building are fairly well described in the Materials and methods section and [Supplementary-material SD1-data], with the exception that the manner in which the CXMS data was enlisted to justify the assignment of ambiguous densities was not documented or explained at all. This information could be added to [Supplementary-material SD1-data] to indicate which crosslinks tabulated in the Supplementary Dataset 1 were used to justify each assignment. Similarly, the Supplementary Dataset 1 should be annotated to indicate whether each crosslink is consistent with the final structure and if it was used to make any assignments of density. [Supplementary-material SD1-data] could also be expanded to include published interactions of each AF that were taken into account in density assignments. At a maximal resolution of 4.5A, it can be difficult to trace the proteins and so it is imperative that the authors thoroughly document everything that went into each assignment so that at least readers can judge how convincing each assignment is.*

[Supplementary-material SD1-data] has been supplied with key crosslinks used in assignment. Dataset 1 includes new sheets where the intermolecular crosslinks are annotated.

*It also seems important that the pdb file be deposited in RSCB for every novel crystal structure being cited here and employed in density assignments.*

Every novel crystal structure has been deposited.

*Reviewer #2:*

*In this manuscript the authors use 3D cryo-EM to reconstruct a structure for a pre-40S subunit isolated from yeast lacking the TRAMP component Mtr4. The manuscript presents an exciting structure and makes several bold, and provocative claims.*

*I have several comments and concerns, some simply aimed at a better presentation of the data, others more substantial. I believe these changes will improve the impact of the manuscript, especially given that a similar structure has already been published.*

*1) I believe the FSC curves give an overestimate for the resolution of the dMtr4 complex, due to anisotropy in the sample, which is reflected in the bump at the bottom. This is of course also reflected in the so map in Figure 1. The authors should state a resolution range for each of their maps, as well as an average resolution. In addition they should show the res map for both the front and the back side of the molecules in Figure 1.*

We have indicated the resolution ranges and an average resolution for each map and added the front side view in Figure 1.

*2) This comment leads to the most important one: The manuscript makes very provocative claims, most importantly about the base pairing of U3. It is impossible to evaluate this claim without having access to the map. This is a general comment for EM papers. Reviewers need to have access to the data, because they otherwise cannot evaluate them properly (we get access to biochemical data, sequencing data, why not structural data?). E.g. in the Kornprobst paper the authors incorrectly assigned Kre33. This is obvious once you look into the density (as the density is not symmetrical), but not necessarily from the manuscript. In any case, without access to the map it is impossible to judge whether the authors correctly assigned the helices base pairing with U3.*

We have provided the cyro-EM maps and the pdb file of 90S for your assessment.

*3) Along the same lines, the authors make a number of surprising assignments, including that of H44 at a very different location. This would be a neat finding, but again is hard to judge without a map. Even more surprising is the assignment of the D-site. I am sure the map provides confidence in a stem-loop. However, there are hundreds of stemloops in this molecule. How the authors decide it is a stemloop at the D-site, which has been essentially disproven to exist is totally unclear.*

The D-site hairpin was assigned mainly based on its binding to Pno1, which has been assigned confidently. In addition, the size of modeled hairpin is consistent with the predicted D-site hairpin and the position of D-site hairpin is close to the base of helix 44.

*4) What is the RNA in the molecule? The authors should show a Northern blot.*

Northern blot analysis has been provided in Figure 1.

*5) The authors use a lot of "unpublished crystal structures" to fit their structure. These structures need to be shown as supplemental data, and pdb files need to be submitted and referenced herein. Examples are: the Utp10 structure, Enp1 structure, the Tsr1 structure, the RRp5-TPR structure, the Imp3 and Rpf1/Rrs1 structure.*

These structures have been deposited and referenced.

*6) Subsection “U3 snoRNP”: "The maps were essentially superimposable." This claim needs a Figure.*

Alignment of three maps is shown in Figure 1—figure supplement 6.

*7) Imp3 and Imp4 are in two different Figures and it is unclear how they relate to each other. Correll and co-workers have shown that they function together. Thus, they should be shown together. There should also be references to that work. As well as previous work from the Baserga lab, which predicted U3 and 18S interactions with Imp3 and Imp4.*

Imp3 and Imp4 are shown together in Figure 7. They are linked by Mpp10 and close to each other. As Imp4 does not interact with 5' ETS and U3 RNA, it is discussed in the 3' major domain section. The related references by Correll and Baserga are cited.

*Presentation issues:*

*1) The overview Figure 2 is presented as done by Kornprobst. This Figure, in my mind was presented really poorly, and I would strongly urge the authors to give a different view. I realize that this will be effort as they need to remake some Figures, but I think it will lead to most people reading this paper and not the Kornprobst paper. Specifically, I would like to suggest that they orient the complex such that the 5'-domain is in the canonical view from the subunit interface (as in Figure 8), where it is at the bottom and not at 2 o'clock (on an imaginary clock). This would also require a rewrite of the overall structure description. I really think it will be worthwhile as it will help people to think about the molecule.*

We understand this reviewer's concern. As the 90S structure is 4 times larger than and dramatically different from the mature 40S subunit, the canonical view of 40S may not be the best way to show the features of 90S. We would like to keep the current orientation unchanged as it guides our presentation of all structures and maps.

2) Of course this would also require different view for Figure 3, Figure 4, Figure 8

See above.

*3) In addition, I would strongly urge the authors to include a panel in most of the other Figures to help the reader orient him/herself with respect to that view (Figure 4, Figure 5, Figure 7, Figure 9–Figure 11. This would be an overview Figure, its turn to achieve the presented view, and the zoombox in that turned view.*

Overview figures were included in many panels to show the displayed regions in the 90S map.

*Reviewer #3:*

*The authors report an impressive structural analysis of early pre-ribosome structure, combining cryo-EM and CXMS. The work provides numerous insights and will doubtless inform many future studies. I have only minor suggestions for improvement.*

*Specific points:*

*1) Noc4-TAP was purified from cells at OD600 = 2.5-3.5 and Dhr1 and Mtr4 depletions from cells 0.8-1.0. Ribosome biogenesis, especially co-transcriptional processing, is known to depend on growth phase. Do the authors see any important differences between Noc4-TAP and Enp1-TAP Dhr1 depl (or Mtr4 depl) models that might reflect the preponderance of cotranscriptional vs. posttranscriptional cleavage?*

On depletion of Dhr1 or Mtr4, 90S strongly accumulated and the A1 cleavage is inhibited. The differences between the three samples should be mainly caused by the depletion of Dhr1 and Mtr4.

*2) Subsection “Structure of 5' ETS and U3 RNA”: H5, H10 and its flanking sequences to the A0 and A1 sites were not located. Is it possible that Enp1-TAP Dhr1 depleted particles are dead-end intermediates in which A0 and A1 cleavage has already occurred? Northern blot analysis of the RNA content of these particles would have been useful.*

We have conducted northern blot analysis for the three 90S samples (Figure 1). The 90S samples from the ΔMtr4 or ΔDhr1 yeast contain highly abundant 22S pre-rRNA, suggesting that the A0 cleavage is normal but the A1 cleavage is inhibited.

*3) Figure 8—figure supplement 1: Structure probing previously showed that there is no secondary structure of RNA at site D.*

Structural probing probably mainly detected the 20S pre-rRNA in the more abundant pre-40S particle where the D site region is single-stranded.

*4) Could the authors discuss Pno1 binding to D site shown on Figure 1—figure supplement 5 in relation to the finding from Zhang et al. (2016)? Also, Pno1 binding is in agreement with previous cross-linking experiments.*

During the stepwise assembly process of 90S, Pno1 is bound after the transcription of helix 44 and before the emergence of the D site sequence, suggesting that Pno1 is initially recruited by the protein-protein interaction with the UTPB complex. The crosslinking between Pno1 and the D site RNA has been added as evidence supporting the assignment.

*5) The potential roles of Utp24 and Rcl1 in pre-rRNA cleavage has recently been debated. Although the structure lacks the actual A1 and A2 cleavage sites, can the authors usefully comment on the positions of Rcl1-Bms1 and Utp24 relative to these sites?*

Utp24 is located near the U3-18S duplex Hc, supporting its role as the A1 site nuclease. As the ITS1 sequence (212 nt) between the A2 and D sites likely folds into a long hairpin, the A2 and D sites may be close to each other in space. In the 90S structure, the D site is separated ~130 Å from Rcl1 by the tandem WD domains of Utp12 and Utp13. This structural consideration does not support that Rcl1 is close to the A2 site.

*Reviewer #4:*

*[…] I only have one major concern:*

*Ye et al. argue that some of the assignment differences relatively to Kornprobst et al. could be due to species-specificity although the 90S appears very well conserved. While politically correct, I find this statement a bit weak and invite Ye et al. to provide arguments and to discuss the chosen contrasted assignments compared to those by Kornprobst et al. Indeed it is capital that the reader understands their assignment and whether Ye et al. believes that their interpretation is more accurate and why! Otherwise the assignment appears only arbitrary and a matter of personal preference.*

We are discussing two kinds of differences, in the density map and the model. We stated "These differences in density map may be caused by species-specific features, different maturation stages of purified 90S and different sample preparation procedures." Some species-specific differences do exist. For example, in the Ct UTPA complex, Utp9 is absent and Utp5 lacks a WD domain.

Our model is based on cryo-EM maps of better resolution and completeness, additional high-resolution crystal structures and chemical crosslinking data and is of much higher quality over the Ct model. Nevertheless, some assignments, especially those without high-resolution crystal structures, should be considered tentative at the current resolution of cryo-EM map.

[Editors' note: further revisions were requested prior to acceptance, as described below.]

*The manuscript has been substantially improved but there are few remaining issues raised by reviewer #2 that need to be addressed before acceptance, as outlined below:*

*1) (Relating to the following comment in the previous review) "The manuscript makes very provocative claims, most importantly about the base pairing of U3. It is impossible to evaluate this claim without having access to the map." My inspection of the map indicates that it does not support the claim that U3 base pairs with H27. Rather, the map shows a duplex, which the authors have assigned as U3 and H27. However, there is no way of telling that H27 is part of this duplex, because there is no density connecting it to either H26 or H28, and indeed 20 nt are missing on either side. In addition, it is unclear that U3 is the other part of the duplex, because there is no EM density for parts of it. The authors must remove the claim that U3 base pairs to helix 27. Similarly, the map does not support the placement of Mpp10, as it is free floating, and the authors should make the speculative nature of their assignment of Mpp10 helices more clear. As in: "Based on the known interactions with Imp3 and Imp4, we assign these to Mpp10, although they could come from other nearby proteins as well."*

The Hd helix is an important feature of 90S structure and we respectfully disagree with this reviewer's suggestion to remove the assignment. In fact, our assignment of the Hd helix to U3 and H27 was supported by ample evidence and clear logic. 1) The Hc helix, which is adjacent to Hd, can be confidently assigned to nt 15-22 of U3 and nt 9-16 of 18S rRNA. 2) The U3 strand of Hc is connected to one strand of Hd by continuous densities, indicating that the Hd helix must contain nt 1-13 of U3. 3) The length of Hd matches exactly the length of nt 1-13 of U3. 4) The H27 sequence is complementary to nt 1-13 of U3 in an evolutionarily conserved manner and is the best candidate for the other stand of Hd. 5) The modeled H27 sequence is located at a middle position between H26 and H28. This arrangement allows H27 be linked to H26 and H28 by ~20 nt at either sides, although both the linkers are not modeled in the map. 6) All other unassigned duplexes in the 5' ETS, 18S rRNA and ITS1 are located too far away and not of the right size (13 bp) to be considered as Hd. Therefore, the assignment of Hd is scientific and solid.

Two α-helices of Mpp10 were assigned. One helix bound to Imp3 can be confidently assigned based on the crystal structure of the Imp3-Mpp10 complex (PDB 5WXM). As to the helix bound to Imp4, the related sentence has been revised to "A short α-helix bound at the protein-binding pocket of Imp4 was tentatively assigned to Mpp10 based on the known interaction between Imp4 and Mpp10 (Lee and Baserga 1999; Wehner and Baserga 2002)"

*2) (Relating to the following comment in the previous review) "the authors make a number of surprising assignments, including that of H44 at a very different location." In their response to this comment the authors indicate that H44 has been assigned based on its interaction with Pno1, which is assigned confidently. However, Pno1 has been shown to bind single stranded RNA, not double stranded RNA, and Lamanna and Karbstein have shown that D-site is single stranded not in a hairpin. Thus, there is no "predicted" duplex. Reviewer 3 had similar concerns. Moreover, looking at the map, I actually have no faith that this is a hairpin. There is density extending from the loop, which shouldn't be there, if it really were a hairpin. I strongly suggest the authors take out the assignment of the D-site hairpin. Minimally, the authors must present this logic in the paper, and add that the identified density might be a hairpin, and if so, could be the D-site hairpin, although previous data on later intermediates indicate that at that stage D-site is single stranded. In other words, this interpretation rests on assumptions, which are unproven, and probably untrue.*

We assume that this reviewer is talking about the D-site hairpin, not H44. Pno1 contains a degenerated KH domain at the N-terminus and a classic KH domain at the C-terminus. In our model, the degenerated KH domain contacts the duplex part of the D-site hairpin whereas the classic KH domain would bind the 5' single-stranded region adjacent to the hairpin. Therefore, the interaction between Pno1 and the D-site hairpin is still consistent with the known RNA-binding model of KH domain.

As to the D-site hairpin assignment, we have revised the related sentences: "A density that is suggestive of a short RNA hairpin is sandwiched between Pno1 and the WD domain of Utp13 (Figure 1—figure supplement 5). Because the D site RNA potentially forms a short hairpin (Figure 8—figure supplement 1) and binds Pno1, the density was temporarily assigned to the D site RNA. Although the D site RNA was found by chemical probing to be single-stranded (Lamanna and Karbstein 2009), this result may mainly reflect the structure of pre-rRNA in the more abundant pre-40S ribosome."

*3) (Relating to the following comment in the previous review) "What is the RNA in the molecule? The authors should show a Northern blot." Interestingly, even though the majority of the RNA is 22S, the model still has the 5'-ETS. Thus, either it remains bound through non-covalent interactions, or the EM preferentially shows a minor population. The authors must discuss this.*

The cleaved 5'-A0 fragment of 5' ETS indeed still binds 90S, as the Northern blot shows that the 5'-A0 fragment has a comparable amount to the U3 snoRNA in the Noc4-TAP and ΔMtr4 particles (Figure 1, probe 5' -A0). This has been discussed in the revised manuscript Results section paragraph four.

*4) Discussion section: Osheim did not show that transcription of H44 brings by a compaction event; Osheim showed that there is compaction after transcription of nearly all of ITS1, and that cleavage requires transcription of 25S.*

Analysis of the assembly of 3'-truncated pre-rRNA fragments shows that transcription of helix 44 triggers the assembly of a mature 90S, but this approach does not consider the kinetics in pre-ribosome assembly. The chromatin spread experiment by Osheim et al. provides a real-time image of pre-ribosome folding. The delayed appearance of 90S in the chromatin spread can be accounted for by the time taken to compact 90S.